# A non-stationary extreme value approach for climate projection ensembles: application to snow loads in the French Alps

Erwan Le Roux[1], Guillaume Evin[1], Nicolas Eckert[1], Juliette Blanchet[2], and Samuel Morin[3]

[1]Univ. Grenoble Alpes, INRAE, UR ETNA, Grenoble, France
[2]Univ. Grenoble Alpes, Grenoble INP, CNRS, IRD, IGE, Grenoble, France
[3]Univ. Grenoble Alpes, Univ. Toulouse, Météo France, CNRS, CNRM, CEN, Grenoble, France

**Correspondence:** Guillaume Evin (guillaume.evin@inrae.fr)

**Abstract.** Anticipating risks related to climate extremes often relies on the quantification of large return levels (values exceeded with small probability) from climate projection ensembles. Current approaches based on multi-model ensembles (MMEs) usually estimate return levels separately for each climate simulation of the MME. By contrast, using MME obtained with different combinations of general circulation model (GCM) and regional climate model (RCM), our approach estimates return levels together from the past observations and all GCM-RCM pairs, considering both historical and future periods. The proposed methodology seeks to provide estimates of projected return levels accounting for the variability of individual GCM-RCM trajectories, with a robust quantification of uncertainties. To this aim, we introduce a flexible non-stationary generalized extreme value (GEV) distribution that includes i) piecewise linear functions to model the changes in the three GEV parameters ii) adjustment coefficients for the location and scale parameters to adjust the GEV distributions of the GCM-RCM pairs with respect to the GEV distribution of the past observations. Our application focuses on snow load at 1500 m elevation for the 23 massifs of the French Alps. Annual maxima are available for 20 adjusted GCM-RCM pairs from the EURO-CORDEX experiment, under the scenario RCP8.5. Our results show with a model-as-truth experiment that at least two linear pieces should be considered for the piecewise linear functions. We also show, with a split-sample experiment, that eight massifs should consider adjustment coefficients. These two experiments help us select the GEV parameterizations for each massif. Finally, using these selected parameterizations, we find that the 50-year return level of snow load is projected to decrease in all massifs, by -2.9 kN m$^{-2}$ (-50%) on average between 1986-2005 and 2080-2099 at 1500 m elevation and RCP8.5. This paper extends the recent idea to constrain climate projection ensembles using past observations to climate extremes.

## 1 Introduction

The use of climate model simulations is the main scientific paradigm to anticipate extreme climate events. In particular, multimodel GCM-RCM ensembles are widely used to quantify the changes in climate extremes and their uncertainties (IPCC, 2021). General circulation models (GCMs) represent key processes of the climate system relevant at the global scale, and provide input for regional climate models (RCMs) used to downscale and refine the climate projections at the local to regional scale.

Climate extremes are often assessed within the statistical framework of extreme value theory (EVT), by focusing either on annual maxima or on values exceeding a high threshold (Coles, 2001). EVT makes it possible to estimate return levels, i.e. extreme quantiles that occur on average once every $T$ years, where $T$ is the corresponding return period. Return levels play a key role in the design of structures (dams, protections, roofs) to withstand the effects of natural hazards (floods, avalanches, wildfires, snow loads), see e.g. Rao and Hamed (2000); Eckert et al. (2008); Evin et al. (2018); Le Roux et al. (2020).

Most approaches using EVT to study climate extremes from multi-model ensembles (MMEs) rely on stationary generalized extreme value (GEV) distributions estimated separately on each climate simulation of the MME, i.e. with each ensemble member (Kharin et al., 2007; Beniston et al., 2007). Specifically, for each ensemble member, annual maxima are assumed stationary for two time periods of 20/30 years: one in the historical period representing the late 20th century climate, and one in the future period. For instance, Fowler et al. (2007) opted for two 30-year time periods: 1961-1990 and 2071-2100, a 30-year time window corresponding to the usual duration which is used to describe the statistical properties of a climate according to World Meteorological Organization (WMO) standards. Next, stationary 20/30-year return levels are estimated for each time period with a GEV distribution. Finally, average changes, i.e. differences of return levels between time periods averaged on all ensemble members, are usually reported (Kharin et al., 2013; O'Gorman, 2014). However, such approaches based on stationary GEV distributions have several drawbacks. First, the assumption of stationarity for 20/30 consecutive annual maxima can be debatable, and the possibility of a trend within the 20/30 years time periods is often not checked (Kharin and Zwiers, 2004). Then, the choice to rely only on 20/30 maxima implies that the estimated GEV parameters have large uncertainties due to the small number of values used. In this case, large return levels, e.g. 50-year (or even larger) return periods which are usually considered to design structures (see e.g., Tab.1 of Cabrera et al., 2012), can be highly uncertain.

Temporal non-stationary GEV approaches address these limitations by taking into account all the available annual maxima for each ensemble member, i.e. all the historical and future annual maxima are fitted with a single statistical model (Kharin and Zwiers, 2004). Such approaches combine a stationary random component (a fixed extreme value distribution) with non-stationary deterministic functions that map each temporal covariate (such as the years or the global mean temperatures) to the changing parameters of the distribution. Another advantage of temporal non-stationary approaches is that they allow return levels to be estimated conditionally on each temporal covariate (Kharin et al., 2013).

A majority of temporal non-stationary approaches for MMEs rely on the GEV distributions estimated separately with each climate simulation of the MME (Tab. 1), with some exceptions (Caires et al., 2006; Kyselý et al., 2010; Roth et al., 2014; Winter et al., 2017), and report return levels (conditionally on a given covariate) averaged on all ensemble members. We believe that such approaches are sub-optimal because they estimate one non-stationary GEV distribution with each climate simulation of the MME, i.e. with roughly less than 200 maxima values, which often implies a simple parameterization (linear) for the non-stationary functions (Tab. 1).

The present study introduces an alternative approach which relies on a temporal non-stationary GEV model fitted to all ensemble members. This approach enables us to quantify uncertainties using standard tools from non-stationary extreme value analysis. Such an approach has mainly been proposed for initial condition ensembles (Tab. 1), i.e. ensemble members that consist of replicates from the same GCM-RCM pair (or same GCM for GCM ensembles) simulated with different initial con-

ditions. For initial condition ensembles, this alternative approach estimates a single non-stationary distribution on all ensemble members by assuming that they are independent and identically distributed (*iid*). However, this alternative approach is inadequate for GCM-RCM ensembles with several GCMs because the *iid* assumption, i.e. that all GCM-RCM pairs follow the same non-stationary distribution, is unlikely to hold in all the cases. Our study fills this gap with a novel non-stationary extreme value approach inspired by the recent trend of statistical methods that constrain climate projections using past observations (Brunner et al., 2020). We propose to fit a non-stationary GEV distribution to past observations and all GCM-RCM pairs, without necessarily assuming that all GCM-RCM pairs follow the same distribution. To this end, we introduce parameters (so-called adjustment coefficients) for the location and scale parameters of the GEV distribution that can account for systematic differences between the different climate trajectories. Different parameterizations of these adjustment coefficients are tested in order to describe the variability between the climate trajectories, the best parameterization being selected using split-sample tests on past observations.

Besides, non-stationary GEV based approaches for climate projections ensembles usually consider linear functions for the non-stationary functions, with the exception of the study by Um et al. (2017) that applies nonlinear functions (Tab. 1). In this study, we extend these approaches by considering piecewise linear functions for the non-stationary functions.

| Ensemble members are fitted | Reference | Adjustment coefficients for the GEV parameters of ensemble members | Non-stationary functions for the GEV parameters | Extreme variable |
| --- | --- | --- | --- | --- |
| Separately | Fowler et al. (2010) | $\times$ | Linear | Precipitation |
| | Hanel and Buishand (2011) | $\times$ | Linear | Precipitation |
| | Kharin et al. (2013) | $\times$ | Linear | Temperature & Precipitation |
| | Brown et al. (2014) | $\checkmark$ | Linear | Temperature & Rainfall |
| | Um et al. (2017) | $\times$ | Nonlinear | Precipitation |
| | Tramblay and Somot (2018) | $\times$ | Linear | Precipitation |
| Together | Kharin and Zwiers (2004) | $\times$* | Linear | Temperature & Precipitation |
| | Wang et al. (2004) | $\times$* | Linear | Significant wave height |
| | Aalbers et al. (2018) | $\times$* | Linear | Precipitation |
| | Fix et al. (2018) | $\times$* | Linear | Precipitation |
| | Wehner (2020) | $\times$ | Linear | Temperature & Precipitation |
| | **Our approach** | $\checkmark$ | Piecewise linear | Snow Load |

**Table 1.** Temporal non-stationary GEV based approaches for GCM ensembles and GCM-RCM ensembles. The symbol "*" means that the ensemble is an initial condition ensemble, i.e. each ensemble member consists of the same GCM-RCM pair with different initialisations.

We illustrate the proposed methodology with an application to snow load data, which corresponds to the pressure exerted by accumulated snow on the ground (proportional to the snow water equivalent). The probabilistic assessment of ground snow load is of major interest for the structural design of buildings (Croce et al., 2018), water resource management (Marty et al., 2017),

or for the prevention of large-scale environmental or infrastructure damages caused by snow storms (e.g. damages to forests, transportation networks, electricity networks). Since annual means of snow loads are expected to decrease under future climate change, it could be expected that extreme snow load would also decrease. However, in cold regions (high elevation regions for instance) extreme snowfall is expected to increase with climate change (O'Gorman, 2014), and this increase of extreme

snowfall can possibly lead to an increase of extreme snow load. This application verifies if snow load extremes are expected to decrease in the French Alps, a quantification of these decreases being of prime interest to study compounds extremes, e.g. extreme snow load combined with extreme wind, or to adapt structures standards, e.g. to decrease the constraints used to design new structures (Le Roux et al., 2020).

Section 2 presents our data, i.e. the 20 GCM-RCM pairs for RCP8.5 adjusted from EURO-CORDEX, and the S2M reanalysis

set as the reference observational dataset (Vernay et al., 2019, 2022). In Section 3, we detail our statistical methodology. Finally, results, discussions and conclusions are introduced in Sects. 4, 5 and 6 respectively.

## 2 Data

Our application focuses on snow loads at 1500 m elevation in the 23 massifs of the French Alps, i.e. between Lake Geneva to the north and the Mediterranean Sea to the south (Fig. 1). This region, home to the largest ski resorts in the world, is prone to
90 snow-related hazards such as avalanches (Favier et al., 2016; Dkengne Sielenou et al., 2021) which are heavily impacted by ongoing warming (Eckert et al., 2013; Castebrunet et al., 2014). This study estimates potential changes in snow load (i.e. the pressure exerted by the snow) hazard for a high emission scenario (RCP8.5) as a case study, although it could also be applied to other scenarios and variables. Snow load can be defined as the gravitational acceleration ($g = 9.81$ m s$^{-2}$) times the snow water equivalent (in kg m$^{-2}$) and has often the units of kN/m$^2$. Snow water equivalent corresponds to the mass of snow per
95 unit surface area, which also corresponds to the observed depth of accumulated snow (in $m$) multiplied by the snow density (in kg m$^{-3}$). Following the block maxima approach to estimate the hazard of snow load (Sect. 3.1), we compute annual maxima of daily snow load at 1500 m centred on the winter season, e.g. an annual maximum for the year 1959 is the maximum from the 1st of August 1958 to the 31st of July 1959 (Fig. 1).

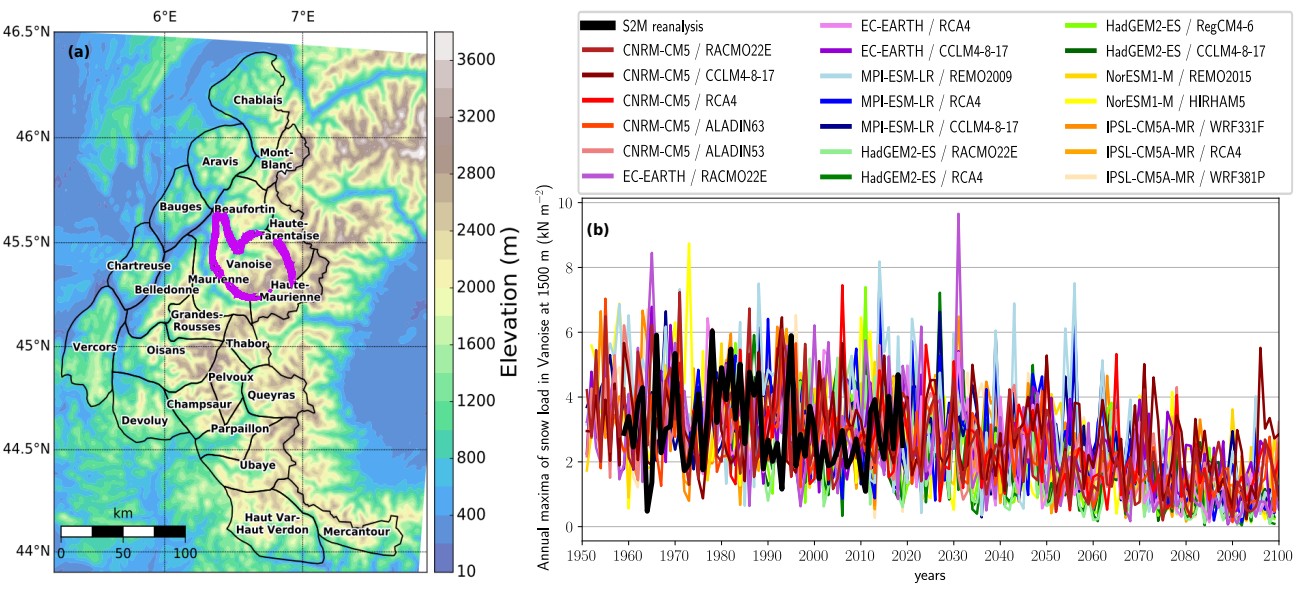

**Figure 1. (a)** Topography and delineation for the 23 massifs of the French Alps, e.g. the Vanoise massif corresponds to the purple region (Durand et al., 2009). **(b)** Time series of annual maxima of daily snow load from 1951 to 2100 for the Vanoise massif at 1500 m elevation. Annual maxima from the S2M reanalysis (1959-2019) are displayed in black, while annual maxima from the 20 adjusted GCM-RCM pairs (1951-2100) under a historical and a high emission scenario (RCP8.5) are displayed with brighter colors.

The S2M reanalysis (Durand et al., 2009; Vernay et al., 2019, 2022) combines large scale reanalyses and forecasts with
100 in situ meteorological observations to provide daily values of snow load from 1959 to 2019. The S2M reanalysis has been both evaluated with in situ temperature and precipitation observations (Durand et al., 2009) and with various snow depth observations (Vionnet et al., 2016; Quéno et al., 2016; Revuelto et al., 2018; Vionnet et al., 2019; Vernay et al., 2022). The

S2M reanalysis focuses on the elevation dependency of meteorological conditions. Indeed, this reanalysis is not produced on a regular grid, but provides data for each massif every 300 m of elevation between 600 m and 3600 m.

ADAMONT (Verfaillie et al., 2017) is a bias-correction and downscaling method which aims at adjusting daily climate projections from a regional climate model against a regional reanalysis of hourly meteorological conditions using quantile mapping. This method was used to adjust the EURO-CORDEX dataset (Jacob et al., 2014) against the S2M reanalysis to provide daily values of snow load that spans historical (1951–2005) and future (2006–2100) time periods. Specifically, the EURO-CORDEX dataset consists of RCMs forced over Europe by GCMs from the CMIP5 ensemble (Taylor et al., 2012) for
the historical and several representative concentration pathways (RCP) scenarios (Moss et al., 2010). We focus on the RCP8.5 emission scenario, and consider a total of 20 GCM-RCM pairs, with 6 GCMs and 11 RCMs (see Supplement, Tab. S1). Finally, every 300 m of elevation for each massif, adjusted EURO-CORDEX meteorological data are used as input to the snow cover model Crocus (Vionnet et al., 2012). This provides estimates of the time evolution of the snow cover (Verfaillie et al., 2018), enabling us to compute the maximum annual value of snow load at 1500 m elevation. For simplicity, we often refer to the S2M
reanalysis as our observation reference. We note that we discard the two most southern massifs because many projected annual maxima are equal to zero.

    The anomaly of global mean surface temperature (GMST) w.r.t. the pre-industrial period (1850-1900) is chosen as the temporal covariate for our statistical methodology. In practice, we smooth this anomaly with cubic splines to obtain a covariate that does not depend on the internal variability of GMST (Fig. 2). For each GCM-RCM pair we rely on the GMST correspond-
ing GCM as covariate, while we rely on GMST from HadCRUT5 (Morice et al., 2021) as covariate for the observations. For simplicity, we refer to +1 °C of smoothed anomaly of GMST as +1 degree of global warming.

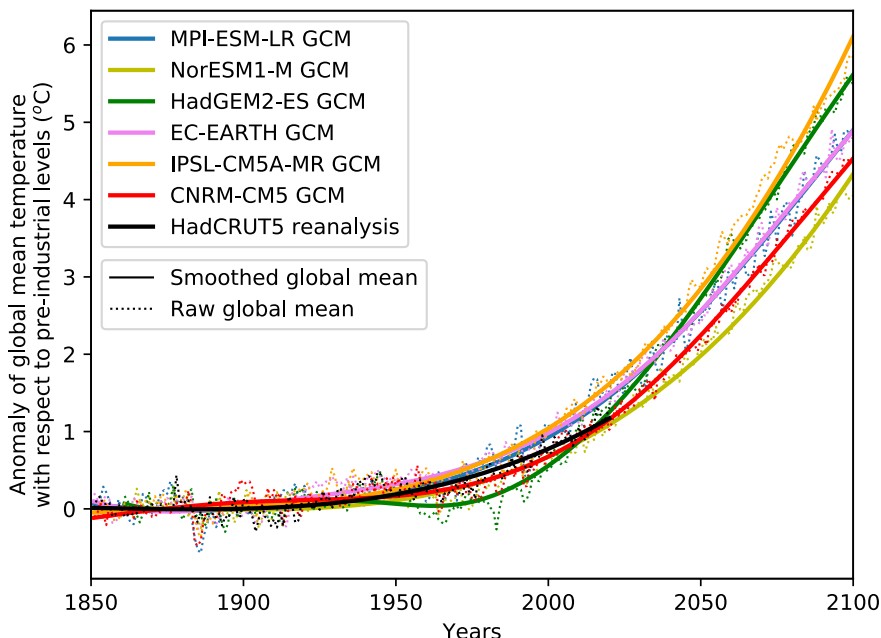

**Figure 2.** Raw output (dotted lines) and smoothed output (plain lines) for the anomaly of global mean annual temperature with respect to industrial levels (1850-1900). For the 6 GCMs, we show the anomaly of global mean surface temperature using historical emissions until 2005, and projected emissions (RCP8.5). Years correspond to periods centered on each winter (Aug-Jul).

## 3  Statistical methodology

### 3.1  Generalized extreme value distribution

Following the block maxima approach of extreme value theory (Coles, 2001), we model annual maxima with the GEV distri-
bution. Indeed, similarly to the central limit theorem that motivates to model means obtained from different samples using a normal distribution, the Fisher–Tippett–Gnedenko theorem (Fisher and Tippett, 1928; Gnedenko, 1943) encourages to model maxima using the GEV distribution. In practice, if $Y$ represents an annual maximum, a natural choice to model the distribution of $Y$ is $Y \sim \text{GEV}(\mu, \sigma, \xi)$, which implies that:

$$P(Y \leq y) = \begin{cases} \exp\left[-(1 + \xi \frac{y-\mu}{\sigma})_+^{-\frac{1}{\xi}}\right] \text{ if } \xi \neq 0 \text{ and where } u_+ \text{ denotes } \max(u, 0), \\ \exp\left[-\exp\left(-\frac{y-\mu}{\sigma}\right)\right] \text{ if } \xi = 0, \end{cases} \tag{1}$$

where the three parameters are: the location $\mu$, the scale $\sigma > 0$, and the shape $\xi$. Three subfamilies of distribution (reversed Weibull, Gumbel, Fréchet) can be derived depending on the sign of the shape parameter ($\xi < 0$, $\xi = 0$, $\xi > 0$) respectively.

In theory, the GEV distribution is adequate when maxima are computed over blocks of infinite size. In practice, the GEV distribution is usually applied to annual maximal and has been shown to provide reliable estimates of return levels in many

hydrometeorological applications (Coles, 2001; Katz et al., 2002; Cooley, 2012; Papalexiou and Koutsoyiannis, 2013). The T-year return level, which is defined as a daily value $y_p$ exceeded each year with probability $p = \frac{1}{T}$, corresponds to the $1 - p$ quantile of the GEV distribution $P(Y \leq y_p) = 1 - p \leftrightarrow y_p = \mu - \frac{\sigma}{\xi}[1 - (-\log(1 - p))^{-\xi}]$. In this study, we set $p = \frac{1}{50}$ as it corresponds to the 50-year return period that is widely used for the design working life of building (Cabrera et al., 2012) notably for the building standard against snow load (Croce et al., 2019).

### 3.2 Non-stationary distribution

Let $Y_t^{obs}$ denote an observed annual maximum for the year $t$ between 1959 and 2019, and $T_t^{obs}$ represent the smoothed anomaly of global mean surface temperature (GMST) from HadCRUT5 for the same year $t$ (Sect. 2). We rely on a non-stationary distribution where each GEV parameter is a piecewise linear function of $T$, the smoothed anomaly of GMST. A log link function for the scale parameter is introduced to ease the numerical optimization.

$$Y_t^{obs}|\boldsymbol{\theta} \sim \text{GEV}(\mu(T_t^{obs}), \sigma(T_t^{obs}), \xi(T_t^{obs})) \quad \text{with} \quad \begin{aligned} \mu(T) &= \mu_0 + \sum_{i=1}^{L} \mu_i \times (T - \kappa_i)_+, \\ \log \sigma(T) &= \sigma_0 + \sum_{i=1}^{L} \sigma_i \times (T - \kappa_i)_+, \\ \xi(T) &= \xi_0 + \sum_{i=1}^{L} \xi_i \times (T - \kappa_i)_+, \end{aligned} \tag{2}$$

where $\boldsymbol{\theta}$ is the vector of parameters $\{\mu_i, \sigma_i, \xi_i, i = 0, \dots, L\}$ for the piecewise-linear functions $\mu(.), \sigma(.), \xi(.), 1 \leq L \leq 4$ corresponds to the number of linear pieces, $\kappa_i = T_{\min} + \frac{(i-1) \times (T_{\max} - T_{\min})}{L}$, and $T_{\min}$ and $T_{\max}$ are the minimum and maximum smoothed anomaly of GMST for the period 1951-2100 (Fig. 2). In other words, $\kappa_2, \dots, \kappa_L$ are fixed and equally spaced between $T_{\min}$ and $T_{\max}$ and correspond to the $L - 1$ anomalies of smoothed GMST where the line breaks, i.e. where the slope of the piecewise linear functions changes.

For a GCM-RCM pair $k$ between 1 and 20, let $Y_t^k$ represent an annual maximum for the year $t$ between 1951 and 2100, and $T_t^k$ represent the smoothed anomaly of GMST (Sect. 2) for the corresponding GCM.

$$Y_t^k|\boldsymbol{\Theta} \sim \text{GEV}(\mu(T_t^k) + \tilde{\mu}_k, \sigma(T_t^k) + \tilde{\sigma}_k, \xi(T_t^k)), \tag{3}$$

where $\boldsymbol{\Theta}$ denotes the set of parameters $\boldsymbol{\theta}$ and of additional parameters $\tilde{\mu}_k$ and $\tilde{\sigma}_k$, and where $\xi(.)$ is given in Eq. 2. The parameters $\tilde{\mu}_k$ and $\tilde{\sigma}_k$ correspond to different adjustment coefficients defined in Table 2. For the $K = 20$ GCM-RCM pairs, these adjustment coefficients are considered for the location and scale parameters and aim at adjusting the distribution of GCM-RCM pairs. Following Brown et al. (2014), we assume that these adjustment coefficients are constant, i.e. the same for historical and future climates. We consider five parameterizations: zero adjustment coefficients, one adjustment coefficient for all GCM-RCM pairs, one for each GCM, one for each RCM, and one for each GCM-RCM pair. Figure 3 illustrates this concept for a fictive ensemble composed of 4 different GCM/RCM pairs with 2 different GCMs and 2 different RCMs. The right column shows how these adjustment coefficients improve the agreement between the different climate simulations with respect to the observations by removing these first-order discrepancies. Obviously, the parameterization with one additional adjustment coefficient for each GCM-RCM pair leads to a better agreement, but at the cost of a much higher number of parameters.

Finally, the size of the entire vector of parameters $\boldsymbol{\Theta}$ is $3 + 3 \times L + S$, corresponding to three parameters for the intercepts $(\mu_0, \sigma_0, \xi_0)$, $3 \times L$ parameters for the piecewise linear functions describing the temporal evolution of the 3 GEV parameters (see Eq. 2), and $S$ parameters corresponding to the adjustment coefficients (see Table 2).

We did not consider adjustment coefficients on the shape parameter because it sometimes leads to prediction failures. This situation can happen when $\xi(T) < 0$, which means that the predictive distribution has an upper bound, and when some future annual maxima lies above this upper bound.

| Parameterization of the adjustment coefficients | Adjustment coefficient for a pair $k$ $\tilde{\mu}_k$ $\tilde{\sigma}_k$ | Number of adjustment coefficients $S$ |
|---|---|---|
| Zero | 0 0 | 0 |
| One for all GCM-RCM pairs | $\tilde{\mu}$ $\tilde{\sigma}$ | 2 |
| One for each GCM | $\tilde{\mu}_g$ $\tilde{\sigma}_g$ | # GCMs |
| One for each RCM | $\tilde{\mu}_r$ $\tilde{\sigma}_r$ | # RCMs |
| One for each GCM-RCM pair | $\tilde{\mu}_k$ $\tilde{\sigma}_k$ | # GCM-RCM pairs |

**Table 2.** The five parameterizations of the adjustment coefficients $\tilde{\mu}_k$ and $\tilde{\sigma}_k$ considered for the location and scale parameters, respectively. When there is only one coefficient for all GCM-RCM pairs $k$, $\tilde{\mu}_k = \tilde{\mu}$ and $\tilde{\sigma}_k = \tilde{\sigma}$ for any pair $k$. When there is one coefficient per GCM (respectively per RCM), $\tilde{\mu}_k = \tilde{\mu}_g$ and $\tilde{\sigma}_k = \tilde{\sigma}_g$ (respectively $\tilde{\mu}_k = \tilde{\mu}_r$ and $\tilde{\sigma}_k = \tilde{\sigma}_r$), where $g$ and $r$ are subscripts referring to the GCM and RCM of the pair $k$, respectively. The notation # refers to the number of elements for the corresponding set.

### 3.3 Maximum likelihood estimation

For each massif, a temporal non-stationary GEV distribution, parameterized by a vector of coefficients $\boldsymbol{\Theta}$, is estimated using the past observations and all GCM-RCM pairs. Let $\boldsymbol{y} = (y_{1959}^{\text{obs}}, \ldots, y_{2019}^{\text{obs}}, y_{1951}^{1}, \ldots, y_{2100}^{1}, \ldots, y_{1951}^{20}, \ldots, y_{2100}^{20})$ represent a vector with all annual maxima of a given massif, i.e. annual maxima from the observations and from the $K = 20$ GCM-RCM pairs (Sect. 2). The maximum likelihood method provides the most likely parameters $\widehat{\boldsymbol{\Theta}}$ with respect to $\boldsymbol{y}$. The maximum likelihood estimator $\widehat{\boldsymbol{\Theta}}$ is obtained from the past observations and all GCM-RCM pairs by maximizing the following likelihood $p(\boldsymbol{y}|\boldsymbol{\Theta})$:

$$\widehat{\boldsymbol{\Theta}} = \operatorname*{argmax}_{\boldsymbol{\Theta}} p(\boldsymbol{y}|\boldsymbol{\Theta}) \text{ where } p(\boldsymbol{y}|\boldsymbol{\Theta}) = \underbrace{\prod_{t=1959}^{2019} p(y_t^{\text{obs}}|\boldsymbol{\theta})}_{\text{past observations}} \times \underbrace{\prod_{t=1951}^{2100} \prod_{k=1}^{20} p(y_t^k|\boldsymbol{\Theta})}_{\text{20 GCM-RCM pairs}}, \text{ with } \begin{array}{l} p(y_t^{\text{obs}}|\boldsymbol{\theta}) = \frac{\partial P(Y_t^{\text{obs}} \leq y_t^{\text{obs}}|\boldsymbol{\theta})}{\partial y_t^{\text{obs}}}, \\ p(y_t^k|\boldsymbol{\Theta}) = \frac{\partial P(Y_t^k \leq y_t^k|\boldsymbol{\Theta})}{\partial y_t^k}. \end{array} \quad (4)$$

### 3.4 Evaluation experiments

Our first evaluation experiment is a model-as-truth experiment, a.k.a. perfect model experiment, which evaluates long-term predictive performances using future projections (Abramowitz et al., 2019). The observations from the S2M reanalysis (Sect. 2) are discarded for this experiment. Instead, a simulation from a GCM-RCM pair is chosen as pseudo-observations. The calibration set contains the "historical" data (1959-2019) of the GCM-RCM pair chosen as pseudo-observations, and the 19

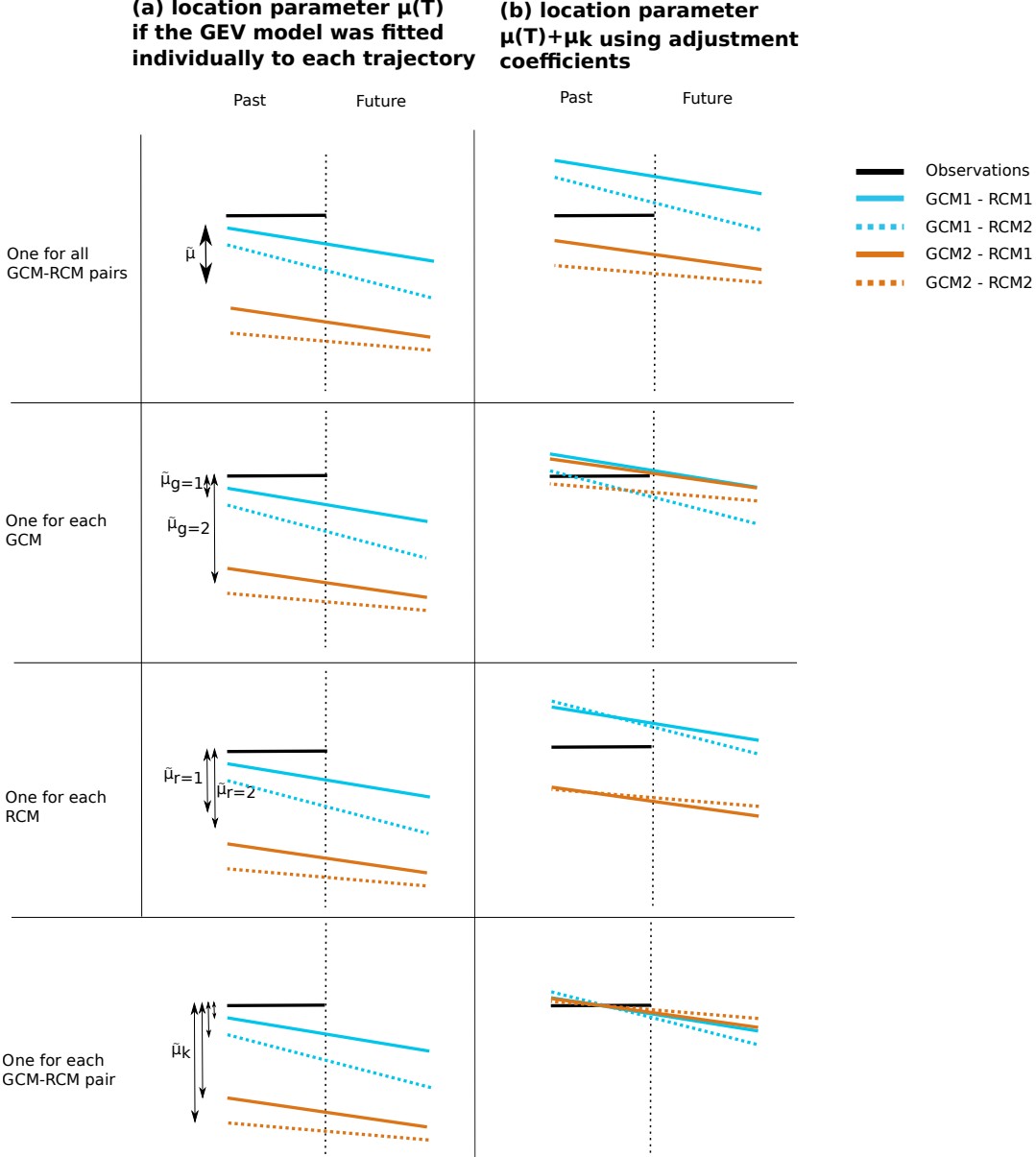

**Figure 3.** Illustration of the evolution of the location parameter $\mu(T)$ on the y-axis as a function of the global warming $T$ on the x-axis for the different options of adjustment coefficients, for a fictive ensemble composed of 4 different GCM/RCM pairs with 2 different GCMs and 2 different RCMs. (a) location parameter $\mu(T)$ if the GEV model was fitted individually to each trajectory. (b) location parameter $\mu(T) + \tilde{\mu}_k$ using adjustment coefficients.

remaining GCM-RCM pairs (1951-2100). The predictive performance is evaluated on an evaluation set that contains the future data (2020-2100) of the GCM-RCM pair chosen as pseudo-observations. In detail, each GCM-RCM pair is successively

regarded as being pseudo-observations. Thus, a model-as-truth experiment can be roughly regarded as a leave-one-out cross-validation w.r.t. to GCM-RCM pairs. We note that for GCM-RCM pairs with the GCM HadGEM2-ES starts in 1982, while the pairs with the RCM RCA4 starts in 1971. Therefore, we successively regard as pseudo-observations the 12 GCM-RCM pairs (out of 20) that start before 1959, i.e. that have annual maxima for the period 1959-2019.

Our second evaluation experiment is a split-sample experiment, a.k.a. calibration–validation experiment, which enables us to estimate the short-term predictive performance of each parameterization. Specifically, for the calibration of the non-stationary GEV distribution, we rely on the oldest observations from the S2M reanalysis (Sect. 2) and all the GCM-RCM pairs. We validate the predictive performance on the most recent observations. For instance, if we choose to keep 80% of the observations for the calibration (1959-2007), then the remaining 20% of the observations are held-out for the evaluation (2008-2019).

In these two evaluation experiments for GCM-RCM ensembles, we calculate the mean logarithmic score ($\overline{\text{LS}}$) on the evaluation set, the lower the better, to assess the out-of-sample skill of a non-stationary distribution parameterized with the parameter set $\hat{\Theta}$ obtained with the calibration set. The logarithmic score is a proper score that can be used to evaluate the predictive performance of the fitted model (Gneiting et al., 2007). For $N$ held-out observations (or pseudo-observations for the model-as-truth experiment) $y^{\text{obs}}_{\text{year}_1}, ..., y^{\text{obs}}_{\text{year}_N}$, we have that $\overline{\text{LS}} = \frac{1}{N} \sum_{n=1}^{N} -\log[p(y^{\text{obs}}_{\text{year}_n}|\hat{\Theta}))]$ where $p(y^{\text{obs}}_t|\hat{\Theta}) = \frac{\partial P(Y^{\text{obs}}_t \leq y^{\text{obs}}_t|\hat{\Theta})}{\partial y^{\text{obs}}_t}$.

## 3.5 Workflow

First, for a set of past and projected annual maxima, we select one parameterization of the GEV distribution (number of linear pieces, parameterization of the adjustment coefficients) using a two-step selection method: i) we select the number of linear pieces with a model-as-truth experiment using zero adjustment coefficients for the GEV parameters ii) we select the parameterization of the adjustment coefficients with three split-sample experiments where the calibration set is composed of $60\%, 70\%$, and $80\%$ of the observations, using the number of linear pieces selected in the model-as-truth experiment. Then, we study trends in the 50-year return level of snow load. For each massif we rely on the parameterization of the GEV distribution selected using the two-step selection method. We report RL50, the 50-year return level that corresponds to Eq. 2, i.e. to the 50-year return level of the observations and their adjusted projections w.r.t. GCM-RCM pairs. In other words, if the selected parameterization has adjustment coefficients, we do not add these coefficients to compute RL50, since adding these coefficients would provide the 50-year return level of the GCM-RCM pairs. The 90% confidence interval is computed using a semi-parametric bootstrap resampling method adapted to non-stationary extreme value distributions (Appendix A). For every anomaly of global mean temperature $T$, we have that the 50-year return level RL50(T) is:

$$\text{RL50}(T) = y_{\frac{1}{50}}(T) = \mu(T) - \frac{\sigma(T)}{\xi(T)}\left[1 - \left(-\log\left(1 - \frac{1}{50}\right)\right)^{-\xi(T)}\right]. \tag{5}$$

## 4 Results

### 4.1 Selection of one parameterization of the GEV distribution for each massif

In Figure 4a, for each massif, we illustrate the selected parameterization of the GEV distribution (number of linear pieces, parameterization of the adjustment coefficients). Among the 23 massifs, Figs. 4b-c highlights the preferred parameterizations after the application of the two-step selection method.

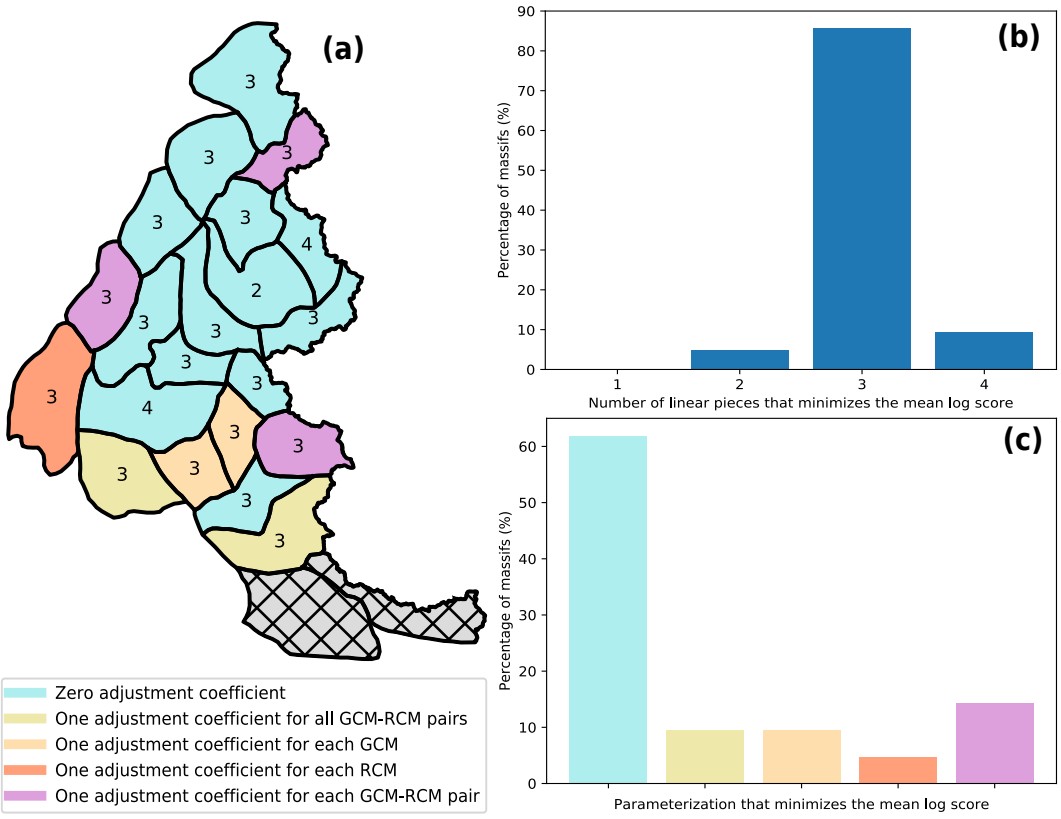

**Figure 4.** **(a)** Map of the selected parameterization of the GEV distribution at 1500 m. The selected parameterizations of the adjustment coefficients are illustrated with colors, while the selected numbers of linear pieces are written on the map. **(b)** Distribution of the selected number of linear pieces. **(c)** Distribution of the selected parameterization of the adjustment coefficients.

In the first step, we select the number of linear pieces that minimizes the mean logarithmic score of a model-as-truth experiment using zero adjustment coefficients for the GEV parameters. The mean logarithmic score is averaged on the held-out pseudo-observations (2020-2100) for each of the 12 GCM-RCM pairs (which are set as pseudo-observations, see Sect. 4.1). We find that the parameterization with three linear pieces minimizes the mean logarithmic score for 80% of the massifs, see Fig. 4b. The parameterization with two linear pieces is selected for one massif, and the one with four linear pieces is selected for two massifs. Thus, at least two linear pieces are selected for the piecewise linear functions. In the second step, we select the

parameterization of the adjustment coefficients (Tab. 2) that minimizes the mean logarithmic score for a split-sample experiment using the number of linear pieces selected in the model-as-truth experiment. The mean logarithmic score is averaged on the evaluation observations for three split-sample experiments, where the calibration set corresponds to $60\%, 70\%$, and $80\%$ of the observations. Indeed, we observe that the split-sample experiment is quite sensitive to the size of the calibration set. Thus, we choose to average the mean logarithmic score on three split-sample experiments to obtain more robust results. We find that the parameterization with zero adjustment coefficients minimizes the mean logarithmic score for two thirds of the massifs, see Fig. 4c. Otherwise, the parameterization with one adjustment coefficient for all GCM-RCM pairs is selected for two massifs, the parameterization with one adjustment coefficient for each GCM is selected for two massifs, the parameterization with one adjustment coefficient for each RCM is selected for one massif, and the parameterization with one adjustment coefficient for each GCM-RCM pair is selected for three massifs. Thus, for two thirds of the massifs, adjustment coefficients do not lead to a better predictive performance on the validation periods. This is presumably due to the fact that GCM-RCM pairs have already been statistically adjusted.

A detailed analysis of the mean logarithmic scores of each parameterization for each massif is provided in the Supplement, Part C.

## 4.2   Trends in the 50-year return level of snow load

In this section, we rely on the parameterization of the GEV distribution selected in Sect. 4.1 for each massif. In Figure 5, we illustrate changes in the 50-year return level between $+1°C$ and $+4°C$ of global warming for four massifs where the selected parameterization is composed of three linear pieces with one adjustment coefficient for all GCM-RCM pairs (Fig. 5a), one coefficient for each GCM (Fig. 5b), one coefficient for each RCM (Fig. 5c), or one coefficient for each GCM-RCM pair (Fig. 5d). In addition, we also perform individual fitting to each GCM/RCM pair, the corresponding return levels being shown with thin gray lines. Note that these estimates are shown for illustrative purposes only, and do not contribute to the final return level estimates indicated with warm colors.

All 50-year return levels (for the non-stationary GEV distribution fitted on the observations, on each GCM-RCM pair, and on the observations and all GCM-RCM pairs) are decreasing with the anomaly of global mean temperature. We observe that RL50 with adjustment coefficients (shown in a warm color) is closer to the 50-year return level of the observation (in dark grey) than RL50 without adjustment coefficients (in cyan). This figure also shows how adjustment coefficients adjust the distribution toward the distribution of the observations by illustrating the probability density functions (with and without adjustment) at $+1$ degree of global warming. Nevertheless, we note that adjusted distributions do not perfectly match the distributions of the observation, which entails that the adjusted RL50 do not match the 50-year return levels of the observation. This is probably because we do not consider adjustment coefficients on the shape parameter. For instance, in Figure 5c, we observe that the shape parameter is negative for the distribution of the observation (because the density has an upper bound), while the shape parameter is positive for the adjusted distribution in orange. We choose to not consider adjustment for the shape parameter because it enables us to constrain predictive distributions on the future period, and to avoid prediction failures (Sect. 5.2). Besides, the 90% confidence interval of RL50 is computed using a semi-parametric bootstrap resampling method adapted to

non-stationary extreme distributions (Appendix A). We note that confidence intervals are widening at the nodes of the piecewise linear functions, i.e. at the anomaly of global temperature where the slope of the GEV parameters changes ($\kappa_i$ in Eq. 2). This is presumably due to the fact that the variability of the three GEV parameters is more important at the nodes than between them.

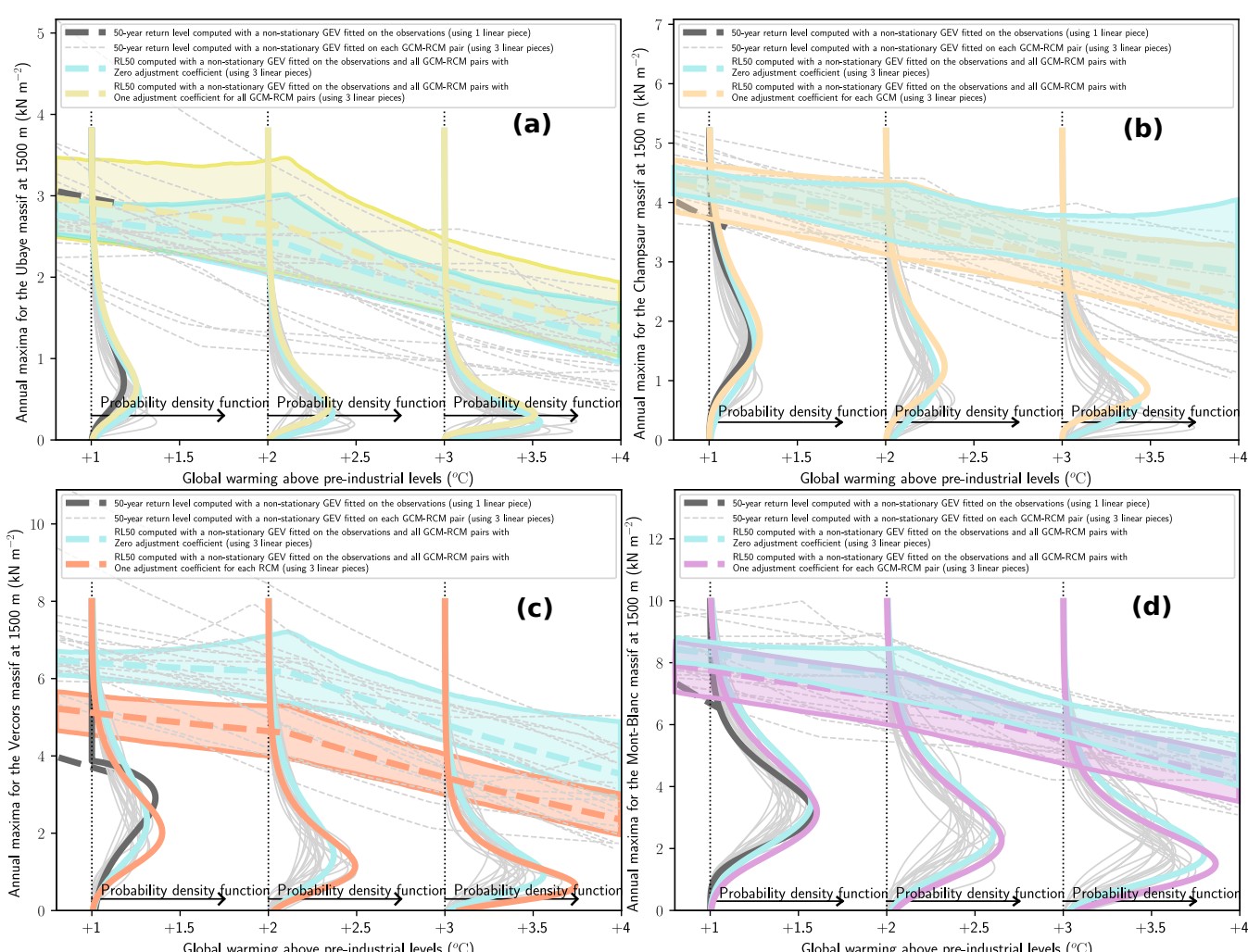

**Figure 5.** Estimated 50-year return levels between +1°C and +4°C of global warming at elevation 1500 m under RCP8.5 for four massifs with different preferred parameterizations for the adjustment coefficients: **(a)** one coefficient for all GCM-RCM pairs **(b)** one coefficient for each GCM, **(c)** one coefficient for each RCM, and **(d)** one coefficient for each GCM-RCM pair. RL50 (Eq. 5) without adjustment coefficients are shown in cyan, and with adjustment coefficients in a warm color. 90% confidence intervals are shaded. The 50-year return levels computed for each GCM-RCM pair (using for each GCM-RCM pair a non-stationary GEV distribution with the selected number of linear pieces) and for the observation (using a non-stationary GEV distribution with one linear piece and a constant shape parameter) are displayed with thin gray lines and thick dark lines, respectively. The probability density functions at +1°C, +2°C and +3°C exemplify how adjustment coefficients can adjust the distribution.

Figure 6 illustrates RL50 for the 23 massifs of the French Alps at 1500 m elevation for +1°C, +2°C, +3°C, and +4°C of global warming, i.e. of smoothed anomaly of global mean surface temperature. The return levels are larger in the northwest

of the French Alps, and this pattern persists with global warming. Over the whole French Alps, the average RL50 equals 5.7 kN m$^{-2}$ at +1°C of global warming, and 3.3 kN m$^{-2}$ at +4°C.

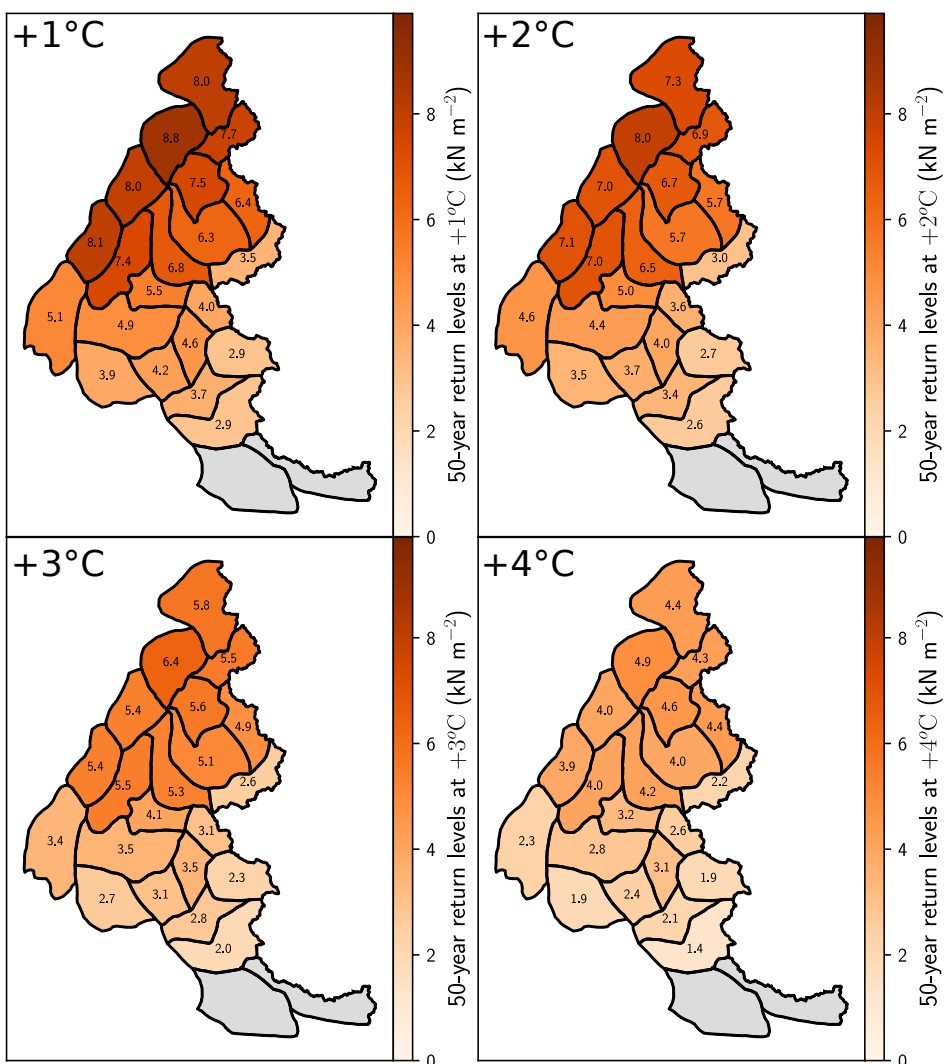

**Figure 6.** 50-year return levels (RL50) of snow load at 1500 m for +1°C, +2°C, +3°C, and +4°C of global warming under RCP8.5.

Figure 7 details the relative change of RL50 for +2°C, +3°C, and +4°C of global warming at 1500 m elevation w.r.t. +1°C, which corresponds roughly to the current level of global warming above industrial levels (see Fig. 2). Over the French Alps, the average change of RL50 is equal to $-0.6$ kN m$^{-2}$ ($-10\%$), $-1.5$ kN m$^{-2}$ ($-27\%$), $-2.5$ kN m$^{-2}$ ($-43\%$) for +2°C, +3°C, and +4°C of global warming, respectively. These relative changes are different for other elevations, a smaller relative decrease being obtained at 2100 m of elevation, and a larger relative decrease at 900 m of elevation (see Supplement, Part B). This result is consistent with the literature (Fig. 2.3 of Hock et al. 2022). At 1500 m, the relative decrease is lower in the center east side

of the French Alps. For instance, for +4°C of global warming, the relative decrease roughly ranges between $-33\%$ and $-38\%$ in the center east side, while it ranges between $-40\%$ and $-54\%$ in the rest of the French Alps.

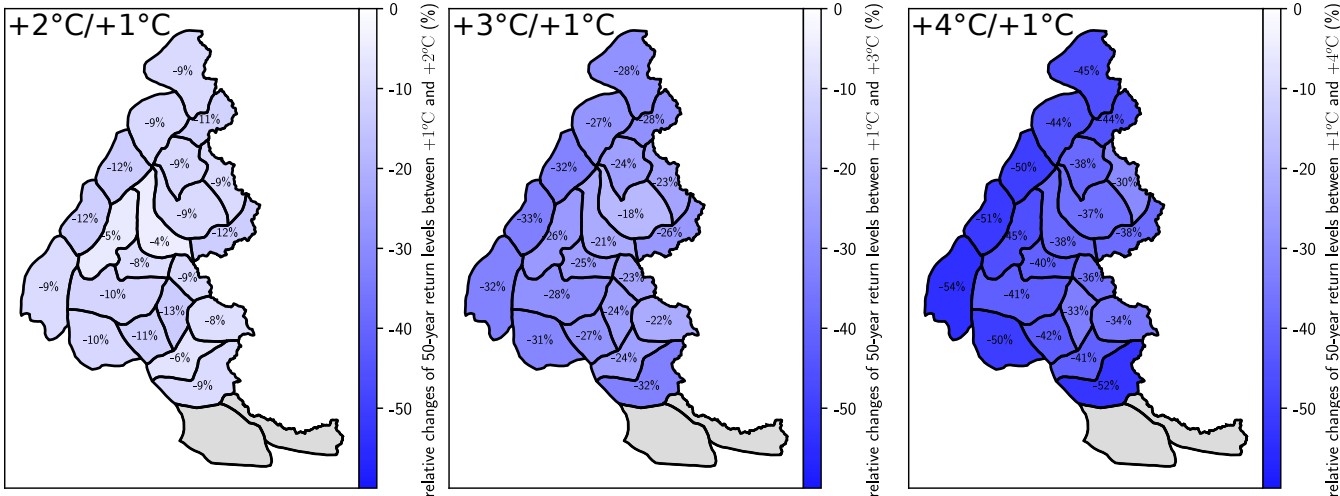

**Figure 7.** Relative changes in 50-year return levels (RL50) of snow load at 1500 m for +2, +3, and +4°C of global warming under the scenario RCP8.5 w.r.t +1°C of global warming.

For each massif, it is also possible to compute the average 50-year return level for several time periods: 1986-2005, 2031-2050, and 2080-2099. For instance, for the time period 1986-2005, the average return level equals the average of the return level found for the years 1986, 1987, ..., 2005. In order to compute the return level of a given year, e.g. 1986, we rely on the
275 relationship between the anomaly of global mean surface temperature (GMST) and the years (Fig. 2). Specifically, we rely on the anomaly of GMST averaged on the six GCMs to compute this relationship. Following this method, we find that on average the 50-year return level is projected to decrease by -0.8 kN m$^{-2}$ (-14%) between 1986-2005 and 2031-2050 and by -2.9 kN m$^{-2}$ (-50%) between 1986-2005 and 2080-2099 under the scenario RCP8.5. Note that this method could also provide the rate of change of other RCPs for various lead times, using their corresponding global temperature values.

**5   Discussion**

**5.1   Comparison of our results with the projected trends at the scale of the European Alps**

In Table 3, we compare our results with the Fig. 2.3[1] of Hock et al. (2022) that provides the trends in winter mean snow water equivalent (SWE) at the scale of the European Alps between 1000 m and 2000 m under the scenario RCP8.5. As detailed in Sect. 2, the snow load is proportional to the SWE, as it is equal to the SWE times the gravitational acceleration (g = 9.81 m s$^{-2}$).
For the 23 massifs, the average return level for several time periods 1986-2005, 2031-2050, 2080-2099 can be obtained as explained in Sect. 4.2. Likewise, with a similar methodology, the mean annual maxima can be expressed as the expectation of

---

[1]https://www.ipcc.ch/srocc/chapter/chapter-2/2-1introduction/ipcc-srocc-ch_2_3/

| Source | Variable | Indicator | Location | Reference period | Future period | Trend |
|--------|----------|-----------|----------|------------------|---------------|-------|
| Hock et al. (2022) | SWE | Mean (Dec to May) | European Alps, 1000-1500 m | 1986-2005 | 2031-2050 | ≈ -35% |
| | | | | | 2080-2099 | ≈ -75% |
| | | | European Alps, 1500-2000 m | | 2031-2050 | ≈ -25% |
| | | | | | 2080-2099 | ≈ -70% |
| **Our results** | Snow Load | Mean annual maxima | French Alps, 1500 m | 1986-2005 | 2031-2050 | -30% |
| | | | | | 2080-2099 | -69% |
| | | 50-year return level | French Alps, 1500 m | 1986-2005 | 2031-2050 | -14% |
| | | | | | 2080-2099 | -50% |

**Table 3.** Projected trends in snow water equivalent (SWE), and snow load under the scenario RCP8.5 using the EURO-CORDEX experiment. In the first four rows of the Table, we specify that the result is approximated because the trend was read from the Figure 2.3. of Hock et al. (2022).

the non-stationary GEV distribution averaged for each year of the time periods. We find a decrease of mean annual maxima of snow load by -30% and -69% for the future periods 2031-2050 and 2080-2099 compared to the reference period 1986-2005.

Figure 2.3 of Hock et al. (2022) relies on the raw (without adjustment) EURO-CORDEX data. They also find decreasing trends. For instance, between 1500 m and 2000 m of elevation, the mean winter SWE (proportional to the mean winter snow load) is expected to approximately decrease by -25% and -70% for the periods 2031-2050 and 2080-2099, respectively (Tab. 3). We observe that our mean annual maxima of snow load has a decreasing rate comparable to the decreasing rate of the mean value of snow load. These comparable rates may stem from the fact that i) both approaches rely (directly or indirectly) on the EURO-CORDEX data, ii) the annual maxima of snow load results from an accumulation during the winter (Dec to May), which implies that we can expect that the mean value will roughly decrease with the same rate as the mean annual maxima.

### 5.2 Methodological choices, assumptions and limitations

For the non-stationarity of the GEV parameters, we choose piecewise linear functions because they can approximate more complex functions with few parameters. This makes our methodology widely applicable. One limitation is that the nodes of the piecewise linear functions are fixed. Yet, we are confident that these functions are well-estimated owing to the large number of maxima: each of the $K = 20$ GCM-RCM pairs provides more than 100 maxima. Otherwise, we rely on the anomaly of global mean temperature as covariate (Sect. 2), like a majority of references cited in Table 1. Indeed, this anomaly is often thought as a good proxy to measure the level of climate change (Fix et al., 2018) which helps strengthen the global response to this threat (Masson-Delmotte et al., 2018). We choose to focus on the scenario RCP8.5 to have the broadest spectrum of potential changes for the 50-year return level of snow load. Also, to obtain Eq. 4 we assume that all annual maxima are conditionally independent given the vector of parameters $\Theta$ which is a classical hypothesis. Following the principle of parsimony, we assume that the adjustment coefficients are constant, i.e. the same for historical and future climates. Besides, as mentioned in Section 3.2, we did not consider adjustment coefficients for the shape parameter because it sometimes leads to prediction failure, i.e.

the predictive distribution gives a null probability to some future annual maxima. It can happen when $\xi(T) < 0$, which means that the predictive distribution has an upper bound, and when some future annual maxima lies above this upper bound. This illustrates the trade-off between i) matching the distribution on the past period with respect to the available observations and ii) having assumptions that help to constrain the predictive distribution on the future period.

For the two-step selection method, we first rely on a model-as-truth experiment to select the number of linear pieces. It assesses the optimal number of linear pieces to predict annual maxima of the pseudo-observations for the evaluation set (2020-2100), i.e. to find a good trade-off between underfitting and overfitting for the calibration set. In this first step, adjustment coefficients are not considered, such that this experiment does not depend on a specific parameterization.

Then, the best parameterization of the adjustment coefficients is selected with a split-sample experiment. It assesses whether applying adjustment coefficients helps to predict observations of the evaluation set, i.e. whether it is reasonable to assume that the observations do not follow the same distribution as the GCM-RCM pairs. The evaluation score is average for three split-sample experiments where the evaluation set corresponds to the last $40\%, 30\%$, and $20\%$ of the observations (Sect. 4.1). Thus, evaluation sets of the three split-sample experiments contain 24, 17, 12 annual maxima, respectively, which is a limited amount of information to select the best parameterization of the adjustment coefficients. As shown in Fig. 4, it tends to favor the most parsimonious parameterization with no adjustment coefficient. Observed maxima have also a limited effect on the estimated parameters since one observed maxima has the same weight than a maxima from a climate model in the likelihood (Eq. (4). As models are fitted to 61 observed maxima and $20 \times 150$ maxima from the different climate models, the selected non-stationary models mostly represent the distribution of the maxima simulated by the climate models. However, it can also be argued that 61 years of past observations has a limited predictive power for long-term horizons where the different trajectories shown by the climate projections can possibly show a great variety of evolution after the observed period. As a comparison, the recent study by Ribes et al. (2021) relies on 170 years of past observations to constrain future climate projections at the global scale. In addition, the proposed methodology has been shown to perform well on temperature projections, but the application to other surface variables (e.g. precipitation, snowfall) needs to be demonstrated.

The 90% confidence intervals of return levels (Fig. 5) account both for the sampling uncertainty (Appendix A) and the climate model uncertainty (distributions are fitted together from the past observations and all GCM-RCM pairs). In contrast, approaches that estimate return levels separately for each ensemble member usually do not account for the sampling uncertainty, i.e. the sampling uncertainty of return levels estimated on each ensemble, even if this uncertainty can be large because return levels are estimated with only one ensemble member. One limitation of our approach is that, contrary to the climatological expectations, the width of confidence intervals does not increase with global warming (Fig. 5). This is presumably a consequence of assuming constant adjustment coefficients.

The goodness-of-fit of the selected models has been tested with the application of the Anderson-Darling test (see Appendix A) to each GCM-RCM pair separately and the observations, for each massif. The test is rejected for 20% of the 483 cases at a significance level of 5% (23 massifs x 21 time series). This relatively high number seems to be mainly explained by the small values reached at the end of the century for many GCM-RCM runs. Indeed, the same tests applied at an elevation of 2700 m show a much smaller percentage of rejections (7%) and larger p-values. The inadequacy of the selected GEV models

to represent these small values can be related to the high proportion of zero snow load values at the end of the century. In these cases, as annual maxima represent maxima of a limited number of positive values, the asymptotic nature of the extreme value theory might be not respected. One alternative could be to consider larger block sizes (maxima over several years). However, the smaller sample sizes would lead to more uncertain parameter estimates. The application of extreme value models to bounded variables thus remains a substantial challenge, especially in a context of climate change where this issue only affects a small part of the dataset.

## 5.3  Related works

First, our methodology based on adjustment coefficients can be seen as an extension of Brown et al. (2014), which estimates non-stationary GEV distribution simultaneously with both observations and a single GCM-RCM pair, and introduces constant bias terms for each GEV parameter. There also exists some links with a debiasing method proposed for annual maxima from GCM-RCM projections (Fontolan et al., 2019). For the location parameter we consider additive adjustment coefficients that can be seen as bias terms, while the adjustment coefficients of the scale parameter that are multiplicative (due to the log link function) can be viewed as bias correction factors (Hosseinzadehtalaei et al., 2021). In this paper, we choose the name "adjustment coefficients" because we introduce them to improve the statistical adjustments. Our idea to add adjustment coefficients for each GCM/RCM or GCM-RCM pairs into the non-stationary extreme value distributions (Tab. 2) comes from the ANOVA framework, which can be applied to partition the uncertainty of GCM-RCM projections by identifying GCM/RCM main effects, or GCM/RCM interactions (Hawkins and Sutton, 2009; Evin et al., 2019, 2021).

Then, our approach based on piecewise linear functions for the non-stationarity of the GEV parameters can be viewed as using linear splines. In the literature, there exists many extreme value theory approaches using splines. For instance, linear splines have been applied to model the temporal non-stationarity (Wilcox et al., 2018), while cubic splines are often considered to model spatial extremes (Chavez-Demoulin and Davison, 2005; Gaume et al., 2013).

## 6  Conclusions and outlooks

Following statistical methods that constrain climate projections using past observations (Brunner et al., 2020; Ribes et al., 2021), we propose a novel approach for GCM-RCM ensembles that aims at fitting a single non-stationary generalized extreme value (GEV) distribution to past observations and to the ensemble of GCM-RCM projections. Specifically, we rely on a non-stationary GEV distribution with i) piecewise linear functions to model the changes in the three GEV parameters ii) adjustment coefficients for the location and scale parameters in order to consider that the GEV model is adequate for the climate projections and the past observations up to systematic shifts for these two GEV parameters. This wide set of GEV models aims at providing a more flexible framework. In particular, piecewise linear functions can represent many possible future changes of the GEV parameters, and include linear trends as special cases.

In order to select the best parameterization of the non-stationary GEV model (number of linear pieces, parameterization of the adjustment coefficients) we design a two-step selection procedure based on two evaluation experiments for GCM-

RCM ensembles: a model-as-truth experiment and a split-sample experiment. The model-as-truth experiment is first applied to select the number of nodes which are required to adequately represent the evolution of the GEV parameters. The split-sample experiment evaluates the added-value provided by the adjustment coefficients, for the different possible parameterizations.

In this article, as a case study, the proposed approach is applied to snow load in the French Alps at 1500 m of elevation, using 20 GCM-RCM pairs statistically adjusted from the EURO-CORDEX experiment under the scenario RCP8.5. More generally, the proposed approach could also be applied to other scenarios, climate variables, and climate projection ensembles. In contrast with most applications of non-stationary GEV models in the literature which consider linear trends, the piecewise linear functions proposed in our approach are well suited to non-monotonic trends.

Many extensions of this work could be considered. First, if adjustment coefficients are not included, our parameterization of the GEV model considers the same non-stationary GEV distribution for the different GCM-RCM pairs. Even in the case where adjustment coefficients are selected, the distributions corresponding to the GCM-RCM pairs are still constrained to have the same changes with global warming because adjustment coefficients are constant. In future works, to better account for different changes of distributions among the GCM-RCM pairs, we could imagine adjustment coefficients that vary with global warming. A second potential extension of this work could be to improve the parameterization of the GEV distribution by adding weights for each GCM-RCM pair. In our methodology, GCM-RCM pairs are currently considered as equally plausible even though it is known that for each application some of them can have a better agreement with the past observations. Following the intuition of weighting schemes for climate ensemble (Knutti et al., 2017), we could design a parameterization of the GEV distribution that assigns more weights, i.e. more confidence, to climate models that agree more with the observations.

Finally, further work is needed to obtain a better agreement between the non-stationary GEV model representing the ensemble of maxima from climate projections and past observed maxima. Indeed, observed maxima are mainly used to identify and correct strong disagreements between the observed and simulated maxima, using adjustment coefficients, and the fitted non-stationary GEV model is not really constrained to represent observed maxima. Bayesian approach representing the predictive distribution of climate projections conditional on historical observations (Robin and Ribes, 2020; Ribes et al., 2021) could be a possible approach to better constrain a GEV model. However, it requires long series of past observations to identify the relationships between the forced climate responses to greenhouse gases obtained from the climate models and from the observations.

**Appendix A: Uncertainty assessment and goodness-of-fit test**

We estimate the uncertainties resulting from in-sample variability with a semi-parametric bootstrap resampling method adapted to non-stationary extreme distributions (Efron and Tibshirani, 1993; Kharin and Zwiers, 2004). This method relies on a transformation $f_{\text{GEV}\rightarrow\text{Standard Gumbel}}$ to the standard Gumbel distribution. Indeed, if $Y_x \sim \text{GEV}(\mu(x), \sigma(x), \xi(x))$, then $f_{\text{GEV}\rightarrow\text{Standard Gumbel}}(Y_x) = \frac{1}{\xi(x)}\log(1+\xi(x)\frac{Y_x-\mu(x)}{\sigma(x)}) \sim \text{Gumbel}(0,1)$. Let $\boldsymbol{y} = (y_1, ..., y_S)$ denote a vector of annual maxima, with $S$ the size of the vector. The transformed variates are computed as $\epsilon_m = f_{\text{GEV}\rightarrow\text{Standard Gumbel}}(y_m)$, using $\widehat{\Theta}$ for $\mu(x), \sigma(x), \xi(x)$.

We generate $B = 1000$ bootstrap samples with a four-step procedure. First, we compute the vector $\boldsymbol{\epsilon} = (\epsilon_1, ..., \epsilon_S)$. Then, for each bootstrap sample $i$, from these transformed variates we draw with replacement a sample of size $S$ denoted as $\tilde{\epsilon}_1^{(i)}, ..., \tilde{\epsilon}_S^{(i)}$. Further, we transform these bootstrapped variates into bootstrapped annual maxima as follows: $\forall m, \tilde{y}_m^{(i)} = f_{\text{GEV} \to \text{Standard Gumbel}}^{-1}(\tilde{\epsilon}_m^{(i)})$.

Finally, we estimate the GEV parameter $\widehat{\boldsymbol{\Theta}}^{(i)}$ with the bootstrapped annual maxima $\tilde{y}_1^i, ..., \tilde{y}_S^i$. To sum up, this bootstrap procedure provides a set $\{\widehat{\boldsymbol{\Theta}}^{(1)}, ..., \widehat{\boldsymbol{\Theta}}^{(i)}, ..., \widehat{\boldsymbol{\Theta}}^{(B)}\}$ of $B$ GEV parameters that represents the in-sample variability.

In addition to the sampling uncertainty, we also assess the goodness-of-fit of the fitted GEV models with the Anderson-Darling statistical test (Abidin et al., 2012). If the non-stationary GEV model is adequate, this test assumes that the transformed variates $\boldsymbol{\epsilon}$ are drawn from a standard Gumbel distribution. Let $F_{emp}$ and $F_{gum}$ denote the cumulative distribution functions of

the empirical and standard Gumbel distributions, respectively. The Anderson-Darling test is based on the following distance:

$$
\begin{aligned}
A^2 &= S \int \Big( F_{emp}(x) - F_{gum}(x) \Big)^2 w(x) \mathrm{d}F_{gum}(x) \\
&\approx -\sum_{i=1}^{S} \frac{2i-1}{S} \Big\{ \log\big[F_{gum}(\epsilon_i)\big] + \log\big[1 - F_{gum}(\epsilon_{S+1-i})\big] \Big\} - S,
\end{aligned}
$$

where $w(x)$ assigns more weight on the tail of the standard Gumbel distribution (see Abidin et al., 2012, for more details).

*Author contributions.*   ELR, GE and NE designed the research. ELR performed the analysis and drafted the first version of the manuscript.
All authors discussed the results and edited the manuscript.

*Competing interests.*   The authors declare that they have no conflict of interest.

*Data availability.*   The full S2M reanalysis on which this study grounds is freely avalaible on AERIS (Vernay et al., 2019). For each GCM, the global mean surface temperature can be computed from https://climexp.knmi.nl/CMIP5/Tglobal/. For the observations, the global mean surface temperature from HadCRUT5 can be downloaded from the following webpage https://crudata.uea.ac.uk/cru/data/temperature/
HadCRUT5.0Analysis_gl.txt.

*Acknowledgements.*   ELR holds a PhD grant from INRAE. We are grateful to Ben Youngman for his "evgam" R package. Inrae, CNRM and IGE are members of Labex OSUG. We are indebted to Raphaëlle Samacoïts from Météo France for providing us the latest version of the climate projection data. The authors also thank the editor, Tamas Bodai, Antonio Speranza, and an anonymous referee, who provided constructive and useful comments.

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
