# Peer review of "A non-stationary extreme value approach for climate projection ensembles: application to snow loads in the French Alps"

_Earth System Dynamics, 2021_

## Referee Comment (RC1)

Reviewer's comment on the manuscript
"A non-stationary extreme value approach for climate projection ensembles: application to snow loads in the French Alps" by Erwan Le Roux et al.

General comments

In this manuscript, the authors show an application of the block maxima approach of Extreme Value Theory (EVT) to annual maxima of snow load for 23 massifs in the French Alps at 1500 m elevation, in a non-stationary setting using a multi model ensemble consisting of General Circulation Models (GCM) and Regional Climate Models (RCM). The aim is to estimate 50-years return levels for the time period 2019-2100. They use a reanalysis data set for the historical period (representing the observations), and a set of RCMs and CMIP5 GCMs. The non-stationarity of this approach relies in using piecewise linear functions linking the evolution in time of the Generalised Extreme Value (GEV) distribution parameters to a smoothed temperature anomaly w.r.t. the pre-industrial period. The estimation is done based on the information provided by the observations and all models together, thus relies on more data than what one would have by considering the models separately. The link between models and observations is based on "adjustment coefficients" which allow the connection between the block maxima statistics of the models and of the observations. The authors test different parametrisations for this adjustment: "zero adjustment", "one for all GCM-RCM pairs", "one for each GCM", "one for each RCM", "one for each GCM-RCM pair". They choose the optimal parametrisation for each massif based on a mean logarithmic score. Similarly, a logarithmic score is used to choose the optimal number of linear pieces for each massif.

The paper raises a very important issue: in order to obtain robust estimates, especially in a non-stationary setting, we need to use a multitude of available information sources. However, it is a challenge to connect properly these different information sources and requires a lot of scientific effort. I consider this paper to be one of the initial steps in the right direction regarding the quantification of changing extreme events due to global warming. The paper has a clear structure, the language is understandable despite the many technical details and steps, the figures are illustrative. I appreciate a lot that the authors discuss critically the drawbacks of their method in Sec. 5.2, and that they provide a clear and concise overview of related studies and how this work is embedded in the current scientific literature. Another strength of this work is that a range of possible parametrisations is explored and the optimal one is chosen in an objective way, based on a mean logarithmic score. I tend to suggest the manuscript to be accepted for publication, but only after a few very important issues are clarified.

I. My main criticism is that the authors use a non-stationary GEV setting without showing that the annual maxima of snow load are properly estimated by a GEV distribution. The authors write that "…due to these theoretical justifications, the GEV distribution enables a robust estimation of return levels". Yes, the GEV distribution enables (under certain conditions) a robust estimation, but only if

1) the chosen block size $n$ is large enough, as the theory applies for $n \to \infty$. The block size of one year is not automatically large enough (one might need 2 years or 10 years or even longer) – it has to be shown that the annual maxima can be reliably estimated based on a GEV distribution (see convergence plots and diagnostic plots in Coles, 2001).

2) the auto-correlation is weak enough - the stronger the auto-correlation, the larger the smallest block size for which the GEV limit is valid. These convergence issues are discussed extensively in Galfi et al. (2017, Complexity).

I ask the authors to show that the convergence to the GEV distribution is good enough for the chosen block size, i.e. that the annual maxima are properly modelled by a GEV distribution. This should be done before building up the non-stationary framework. If this cannot be shown, I'm afraid that the whole experimental setup described in the paper is useless, as it is a necessary condition. In the theoretical description of the GEV distribution in Section 3.1 more emphasize is needed to underline the asymptotic nature of the theory.

III. It is a bit disappointing that even after performing the adjustments and applying the complex methodology, the results still do not follow the observations, as shown by Fig. 4 and discussed shortly in the text. One reason for this could be that, as the authors explain, the evaluation set for adjustment parametrisation contains only a few (24, 17, 10) maxima, thus the selection of the optimal parametrisation might be misleading. I believe a more thorough discussion is needed here. It is also not clear for me how to overcome the issue regarding the estimation of the shape parameter: in case of no adjustment, the model results do not follow the observations, in case it is adjusted it can lead to prediction failures. The authors do not suggest any solution for this, although it would be crucial for the applicability of the method in future studies.

IV. It is not totally clear for me why is it important to know return levels of a variable whose extremes are expected to decrease with global warming (assuming a simple direct relationship between them), thus becoming less extreme? I think the authors should put more emphasize to show the relevance of this subject.

Specific comments

L93 Are the time periods "historical" 1951-2005 and "future" 2006-2100 correct?

L98 It is not mentioned to what "Crocus" refers.

Eq.4 There should be two equations here: $\hat{\theta} = \mathrm{argmax}(p(y|\theta))$ and $p(y|\theta) = \prod \dots$

L177-L178 It is not totally clear what "RL50" stands for because it is written that "is computed without adjustment coefficients", but later in Figure 4 it is used also for the case with adjustment coefficients.

L210 & caption of Fig. 4 GCM-RCM pair instead of RCM-RCM pair

L220 "adjustment" written twice

Figure 4: I think that in case of the line with the "warm" colours the legend should be different for the 4 subplots – the legend is the same for each subplot, although the text and the figure caption suggest the opposite.

Figure 5: To which year or period do these return levels refer?

---

## Referee Comment (RC3)

[referee-annotated manuscript omitted]

---

## Author Comment (AC2)

**Comments from Referee #2**

We thank the reviewer #2 for these comments on our manuscript. Please find below a detailed feedback to individual comments and questions.

Main comments:

RC2#1. The methodological approach proposed in the paper, based on simultaneous use of observations and model projections, is stimulating in particular when facing problems (like the one addressed in the paper) in which numerical simulation models are characterized by heavy tuning-parameterization (fudge factors) changing in time. However, in order to make the estimation process tractable in the specific application considered in the paper, many ad hoc assumptions have to be introduced and the results are admittedly (Section 5.2) problematic, raising doubts concerning the **applicability** of the adopted working hypotheses. This situation often occurs in operational statistical estimation: thorough a posteriori analysis is almost invariably required; the authors should critically re-examine their assumptions.

We thank the reviewer for this comment. We agree that some assumptions are particularly critical and Section 5.2 tries to discuss these limitations. Our ambition for this study is to present different possible nonstationary extreme value models that combine past observations and an ensemble of simulations from climate models. It follows a study by the same authors on past observations (Le Roux et al., 2020) which shows that nonstationary GEV models are adequate for the assessment of ground snow load extremes in the French Alps. We also refer to the comment RC1#1 which presents Goodness-of-Fit tests of the proposed models of this study. It is also important to note that the set of the proposed models is larger that most nonstationary GEV models of the literature. These additional ad-hoc assumptions can actually be viewed as a more general framework. For example, the GEV parameters of nonstationary models often follow linear trends, which is a special case of the piecewise linear functions, i.e. when L=1.

Among the wide range of possible nonstationary GEV models, we have performed many tests in order to select a limited set of applicable models. Some models were overparameterized and could not be fitted (i.e. the parameters could not be identified). This might be a point that could be added to Section 5.2 in order to comfort the proposed approach. On the contrary, it must also be pointed out that modeling the evolution of the GEV parameters using nonlinear functions is particularly challenging, which explains that it is rarely done (see Table 1), whereas monotonic evolutions seem overly simplistic in many applications, especially over a long time period (e.g. more than 150 years). As a consequence, we do not wish to re-examine this particular assumption as it both represents an interesting challenge (how to cope with these nonlinear evolutions) and a necessary framework (for some extreme variables that could reach a "peak" in the future).

We also applied the methodology to other variables (results not shown) and obtained satisfying results in the sense that their interpretations were in agreement with what can be expected from a physical point of view. We are thus confident that the general approach (i.e. the two-step approach presented in Section 3.5 and the use of adjustment coefficients) is promising and can be considered as a solid basis for various applications, although we acknowledge that the current approach has some limitations.

RC2#2. The paper is neat and clean, but here and there not easily legible as it is very concise ("dense"): since the paper proposes issues of potential interest for a wide audience in which "non experienced" readers could find elements of interest I suggest a more

"friendly" communication approach; but I leave to the authors deciding whether being concise is more important than being readily accessible for a wider audience.

It is important for us to reach a wide audience of practitioners and we agree that the current manuscript could be improved from this perspective. For example, a simple sketch illustrating the concept of the "adjustment coefficients" could be added in Section 3.2:

[Figure]

Figure 3: Illustration of the evolution of the location parameter μ(T) for the different options of adjustement coefficients with a fictive ensemble composed of 4 different GCM/RCM pairs with 2 different GCMs and 2 different RCMs. (a) location parameter μ(T) if the GEV model was fitted individually to each trajectory. (b) location parameter μ(T)+$\mu_k$ using adjustement coefficients and a joint estimation.

Specific comments:

RC2#3. Many acronyms and "technical slang" words appear in the paper: a glossary may help.

Thank you for this comment. We define these acronyms when they first appear in the text. If the reviewer thinks that some acronyms or technical terms are not properly defined, we would be pleased to explain them in more detail.

**RC2#4. Line 3** "chain of MME": define in text or in glossary.

MME refers to "multi-model ensembles" as defined in the text at l. 28 and is a standard acronym when a set of simulations obtained from different climate models is used. It is a very common acronym and it does not seem necessary to define it in a glossary in our opinion. The term "chain" will be replaced by "climate simulation". Climate simulations are often referred to as "chains" because it is obtained as the result of a simulation chain: a RCP scenario provides the greenhouse gas concentration trajectories, then a General Circulation Model (GCM) provides large scale climate simulations for these scenarios, then a Regional Climate Model provides climate simulations at a fine spatial resolution forced by the GCM outputs.

**RC2#5. Line 7** "with a robust quantification of uncertainties.": this claim appears repeatedly in the paper; I found mathematical definition in Appendix A: Uncertainty estimation a technical quantification uncertainties, but not an analysis-discussion of the "robustness" of the estimation itself.

We agree that the term "robust" can be used only if there is a clear statistical interpretation. In our case, the quantification of uncertainties assesses in-sample variability using bootstrap methods and we do not provide results about the robustness of these uncertainties. The term "robust" will be removed in the revised version of the manuscript.

**RC2#6. Line 11** "is of major interest for the structural design of roofs": not only (skiing, avalanches, mobility, etc.); a few more words about applications could help.

We agree that snow loads may cause large-scale environmental damages (e.g. to forests) as well as damages to infrastructures (transportation networks, electricity networks) and accidents to people. This will be added to the abstract.

**RC2#7. Line 24** "EVT makes it possible to robustly estimate return levels": see Line 7 comment.

As the term "robust" is not supported by the results, we propose to remove it from the manuscript.

**RC2#8. Line 29** "estimated separately on each chain of the MME": see Line 3 comment.

"Chain" will be replaced by "climate simulation".

**RC2#9. Line 32** "30-year time slices": perhaps it is worth mentioning that 30 years is the traditional (WMO) "time scale" of "climatological" analysis.

Thanks for this comment, this will be added.

**RC2#10. Line 52** "robustly quantify uncertainties": see above lines 7 and 24 comments.

Thank you, see responses above.

**RC2#11. Line 63** "adjustment coefficients": a few more words could help.

Yes, we agree, see our response to the comment RC2#2.

**RC2#12. Line 80** "snow load" see Line 11 comment.

See our response to the comment RC2#6.

RC2#13. **Line 92** "Quantile mapping method ADAMONT": a few words about it?

ADAMONT is a bias-correction and downscaling method which aims at adjusting daily climate projections from a regional climate model against a regional reanalysis of hourly meteorological conditions using quantile mapping. These explanations will be added to the manuscript.

RC2#14. **Line 204** "For a detailed analysis of the mean logarithmic scores of each parameterization for each massif, see Supplement, Part C.": what is Supplement, Part C? Where is it?

Thank you for pointing out this mistake. This supplement was part of a previous version of the manuscript. This sentence will be removed.

RC2#15. **Fig.4** This figure plays a central role in the paper: some graphical features are too faint.

We will try to adjust the colors of the lines/bands in order to improve these graphical features.

RC2#16. **Line 255** "Figure 2.3 of IPCC (2019)": wouldn't it be possible to insert this figure or its direct internet link in the text?

Thank you for this suggestion. The following permalink will be inserted in the revised manuscript:

https://www.ipcc.ch/srocc/chapter/chapter-2/2-1introduction/ipcc-srocc-ch_2_3/

RC2#17. **Line 273** "because it sometimes leads to prediction failure, i.e. the predictive distribution gives a null probability to some future annual maxima.": this is not clear to me!

This situation happens when $\xi(T)<0$, which means that the predictive distribution has an upper bound, and when some future annual maxima lies above this upper bound. This explanation will be added to the text.

RC2#18. **Line 287** "The 90% uncertainty intervals of return levels (Fig. 4) account both for the sampling uncertainty (Appendix A) and the climate model uncertainty (distributions are fitted together from the past observations and all GCM-RCM pairs).": not easy to distinguish in the figure (see comment to Fig.4 above).

We agree that this figure could be improved in order to highlight the differences between the different uncertainty bands. This figure will be modified.

References

Le Roux, E., G. Evin, N. Eckert, J. Blanchet, and S. Morin. « Non-Stationary Extreme Value Analysis of Ground Snow Loads in the French Alps: A Comparison with Building Standards ». *Natural Hazards and Earth System Sciences* 20, n° 11 (6 novembre 2020): 2961-77. https://doi.org/10.5194/nhess-20-2961-2020.

---

## Author Comment (AC3)

We thank Reviewer #3 for these comments and this in-depth review of our manuscript. We appreciate this expertise although we think that some aspects of our work have been misunderstood, probably due to a lack of explanations in the current manuscript. Please find below our responses to these comments.

Main comments:

RC3#1. My feeling is that this paper is not providing a good solution to a real problem. They consider several GCM-RCM model pairs ('model' in the following). In the Introduction the authors point out that previous studies evaluated extreme value statistics (EVS) for individual models and in some cases they took then the ensemble mean of return levels. In this regard the authors are concerned that the estimates for separate models are not so reliable because of data scarcity. However, they seem to do this themselves.

Thank you for this comment. Indeed one of our main motivations for the proposed method is to improve the estimates using a joint estimation approach. Indeed, fitting individual GEV to each climate simulation leads to a set of very uncertain estimates. Taking the mean of very uncertain estimates of a return level is not equivalent to having one estimate of the mean return level obtained from the whole set of climate simulations, as is done in this work. This will be clarified in the revised version of the manuscript.

RC3#2. They introduce the concept of "adjustment coefficients", which is really just a difference of an estimate of the GEV parameter for a particular model (or subset of the data) from the estimate upon lumping all the data. I think we really don't need a name for this, beside the problem that they do what they criticised.

It must be clear that these "adjustement coefficients" are not direct differences between estimates of GEV parameters. They are estimated by maximizing the likelihood function (4) and represent a compromise between fitting past observations and future maxima as obtained from the different climate models. The name for these parameters can of course be debated but they represent the adjustment of a systematic shift between maxima obtained by the different climate models (see Figure 3 above in response to the comment RC2#2). According to the properties of the climate models, this shift can be assumed to be the same for each GCM, each RCM. We also test the possibility of estimating these parameters for all GCM/RCM pairs. In this application, these parameters do not often lead to the best model, as shown in Figure 3c of the submitted manuscript. However, we believe that this possibility can be interesting in other applications where annual maxima from climate models largely differ from observed maxima in the historical period. As indicated in our response to comment RC2#2, a simple sketch of the concept around these parameters will be added in the revised version.

RC3#3. On the other hand, lumping all the data together, in order to have seemingly more robust estimates, is also problematic. As i pointed out in some recent publications of mine, a model ensemble (or multi-model ensemble, MME, as it's most often called) does not represent an objective probability distribution. As such, fitting a GEV to MME data is flawed methodology. It has no meaningful probabilistic interpretation. On the contrary, doing this for (converged) initial condition ensembles is fine. It is actually a profound scientific challenge on how to use MME data in a meaningful way. I don't mean to discourage anyone from trying, though, and hope that real progress can be made. The authors promised a constraining of the estimates/projections using observational data. Emergent constraints is now a popular concept, but it appears to me that the authors did nothing like that. They simply threw the observational data in the mix. However, the information from it is diluted by the large amount of model data.

Thank you for this comment. We entirely agree that fitting probabilistic models to samples obtained from a multi-model ensemble is highly questionable, as it is questionable to apply a statistical test to assess the statistical significance of projected averages obtained from a MME (see Storch and Zwiers, 2013). However, MMEs are widely used to assess the impact of climate change because it seems difficult to trust more one particular model (or one GCM-RCM-impact model chain) than the others. Emerging constraints (e.g. Ribes et al., 2021) try to answer this challenge but we agree that the limitation of our approach must be clearly acknowledged. This will be done in the revised version.

In this work, it is also true that we put the same weight on the observed maxima and the maxima from the climate simulations. As a consequence, it is true that observed maxima have a limited influence on the estimated parameters. Some attempts to put more weights on the observations did not lead to satisfying results because of the limited period covered by the observations, which does not provide much information about the evolution of the GEV parameters. This limitation is also discussed in our response to comment RC1#2 and will be discussed in more depth in the revised manuscript.

RC3#4. Obs data is rather used for bias correction. I'm not sure this was done. Or, if it was done, then it seems to have even less use to throw the obs data into the mix for doing EVS.

A quantile mapping method (ADAMONT) is a bias-correct method that was applied to climate simulations. This method shows some limitations to correct the tail of the distribution since, by definition, extreme values are rarely observed. This explains why some biases can still be expected in the bias-corrected climate simulations.

RC3#5. I share my detailed comments on the manuscript with the authors in an annotated pdf. Hopefully it is useful one way or another. I'm sorry that i cannot be more positive this time. If i misunderstand something, i'm happy to learn from the author's response.

We appreciate these comments and are also happy to discuss them and defend our position.

Specific comments in the annotated pdf

RC3#6. l.3: chain of the MME?

As indicated in our response to comment RC2#4, this term is sometimes used to refer to a particular climate simulation chain (e.g. scenario/GCM/RCM). This term will be removed in the revised manuscript.

RC3#7. l.7: flexible?

"Flexible" was used in the sense that the proposed nonstationary GEV model can be adapted to a great variety of nonlinear evolution thanks to the piecewise functions. This term will be removed since it does not seem to be informative.

RC3#8. l.8 "functions of time"?

As indicated in section 3.2, parameters are functions of anomalies of global mean temperature. This will be added.

RC3#9. l.8. "adjustment coefficients"?

These adjusement coefficients are additional parameters (see our response to the comment RC3#2).

RC3#10. l.17. Maybe this should go in the end of the sentence.

Thank you for this suggestion, this will be done.

RC3#11. l.19: I cannot think anything else.

"One of" will be removed in the revised version.

RC3#12. l.19: Even the most basic climatic means are obtained/projected by simulations.

We agree with the reviewer.

RC3#13. l.23: I would say that quantiles and tail probabilities are more often used by climate scientists than EVT.

It is true that return levels are often obtained using different empirical formulas (e.g. Gringorten) or various distributions. EVT seems to be dominant in the past ten years (see, e.g., Serinaldi and Kilsby, 2014, concerning rainfall extremes). We propose to replace "usually" by "often" to indicate that the EVT approach is undoubtedly popular for the assessment of climate extremes.

RC3#14. l.24. Why robustly? If we are interested in too high return levels, surely the estimates can be very poor.

This comment was also made by reviewer #2 (see comment RC2#7) and we agree that "robustly" should be removed here.

RC3#15. l.29. I don't know what you mean by chain. I guess you mean time series. If so, "chain" is not used in English in this context.

The term "chain" will be removed (see also our response to the comment RC2#4).

RC3#16. l.32. 30 years is not really a slice, is it?

"slices" will be replaced by "periods".

RC3#17. l.44 Do we really need this reference for nonstat EVT? This seems to unnecessarily inflate the bibliography. There is Coles already cited.

This reference will be removed.

RC3#18. l.47 chain

This will be replaced by climate simulation.

RC3#19. l.51 This is poor English. Otherwise, this concept is unfortunately flawed. We can only fit a probability distribution if we have reason to believe that there is an objective physical measure from which our data is drawn from. However, regarding a MME there isn't. It is an ensemble of opportunity (or chance). There isn't a probability space from which Earth System Models are samples. This is what we were articulating in another context of forced response. The right approach is considering Initial condition large ensembles because in that case there is an objective physical probability measure generated by the one climate model that represents some hypothetical physics.

This part of the sentence will be rephrased and replaced by "on a temporal non-stationary GEV model fitted to all ensemble members". See our response to comment RC3#3 concerning the rest of this remark.

RC3#20. l.53 why this reference?

This reference will be removed.

RC3#21. l.56 This should not be assumed but rather checked. What one would check is whether the ensemble still depends on its initialisation or already converged to the snapshot attractor. The check can proceed by checking the stat significance of the distributions (concerning a particular scalar observable) bw TWO ensembles that were initialised differently. (Drotos et al. J Clim 2015).

We agree but it does not seem that convergence has been checked in the papers cited in Table 1 (Kharin and Zwiers (2004), Wang et al. (2004), Aalbers et al. (2018), Fix et al. (2018)).

RC3#22. l.57. On l 51-52 the text reads such that you follow this approach.

Thank you for this comment. This sentence was misleading and will be modified.

RC3#23. l.60 How does this solve the described problem?

Using past observations and adjustment coefficients, this nonstationary GEV model assumes that different climate simulations can be used to project future return levels. This was perhaps unclear from this version of the manuscript and we hope that the proposed amendments will motivate this approach (see also the response to comment RC3#2).

RC3#24. Why do you exclude the shape parameter from this treatment?

This is explained later at l. 137: "because it sometimes leads to prediction failures." for example when a negative shape parameter leads to predictive GEV distribution with an upper bound and when some maxima exceeds this upper bound.

RC3#25. l.63. By this do you mean "the various GEV distributions of all GCM-RCM pairs"? Variability of climate trajectories does not sound anything like that.

Here, we mean that a MME obtained with many different climate models can lead to a large variation of projected changes. This is true for average changes (see, e.g. Evin et al., 2021a for a EURO-CORDEX RCM ensemble) and for extremes (Rajczak and Schär, 2017).

RC3#26. l.64. "jointly fitted" What does this refer to.

This refers to the joint estimation over all simulations of the MME but "jointly" is not really needed here and will be removed.

RC3#27. l.65 Why do you say that past observations represent the most likely climate trajectory? This is so strange language to me. And i'm not convinced that it makes deep sense.

Thanks for this comment. We agree that "to represent the most likely climate trajectory" was not clear, it will be removed.

RC3#28. l.65 By this do you mean that you fit output data from individual models whereby you have different e.g. location parameter estimates for the different models? If so, there

seems to be no difference from what you describe on l46. Rather, you stop short of proceeding with obtaining some ensemble statistics, e.g. mean of the return levels.

Thank you for this comment. As indicated above, the purpose of these adjustment coefficients is to account for systematic shifts in climate simulations, either for the location or the scale parameter. However, we consider different options for these coefficients (one by GCM, one by RCM, etc.) and this will be illustrated by a simple sketch. However, we insist on the fact that these adjustement coefficients are part of the joint inference and are **not** obtained by first fitting individual GEV models to the different climate simulations.

RC3#29. l.68. Does the data look piecewise linear? If not, why not using some polynomial?

Polynomial functions have been tested but can lead to poor models for some types of evolutions. In particular, they are not very well suited to changes of trends (e.g. an increase followed by a decrease). Piecewise linear functions provide a certain flexibility in this regard.

RC3#30. l.76 What does this mean. I suppose you might need to provide a reference here.

This is explained at l.93-94. "Statistically adjusted" will be removed from this sentence.

RC3#31. l.78 resort?

Thank you for this comment, this will be corrected.

RC3#32. It would be good to have a short explanation what it is so that one doesn't need to look it up.

We will add the following definition "the pressure exerted by the snow".

RC3#33. l.85. It would be helpful to say that snow load has the physical unit of pressure, that is Pascal.

Snow load usually has the units of kN/m2. This will be added.

RC3#34. l.85. Otherwise, one has to wonder if this study is very useful in case it is very common to have different snow depths just meters apart, say, from house to house or street to street.

This is a relevant comment and it is true that snow depth is known to vary a lot in space. However, aside from these measurement issues, there is a clear need to assess snow load hazards as it is indicated in the abstract (for the design of roofs) and in our response to the comment RC2#6.

RC3#35. l.85. If i wanted to see what a particular model has in a particular model, i could not find it. Perhaps do not connect the lines.

We agree that it is difficult to look at a particular climate simulation due to the many trajectories. Here the figure mainly aims at showing the overall variability from one year to the other and between the different members of the ensemble.

RC3#36. l.92. "The" is missing.

"The" will be added.

RC3#37. l.92. software package Crocus?

Thank you for this comment. Crocus is actually the part of the S2M numerical chain that produces the evolution of the snowpack. This precision will be added to the manuscript.

This will be added.

Thanks, this will be corrected.

RC3#39. l.101. Is it because this way you have a much better "stat agreement" bw. the models?

It is true that there is a better agreement between the statistical properties of the snowload maxima produced by the different climate models when the covariate is the global temperature than when it is time. Indeed, the climate sensitivity of each GCM can produce large differences of warming for a particular time horizon (e.g. 2050). Taking a warming horizon (e.g. +2°C) removes this "climate sensitivity effect" and makes the different projections more consistent (see also Verfaillie et al., 2018).

RC3#40. l.103. "does not depend on the internal variability of GMST" I don't understand what you mean.

We mean that the "smoothed GMST" is not affected by the internal variability simulated by the GCM (see the fluctuations of the raw GMST at the decadal scale in Fig. 2).

RC3#41. l.109-111: "Indeed theoretically, as the central limit theorem motivates asymptotically sample means modeling with the normal distribution, the Fisher–Tippett–Gnedenko theorem (Fisher and Tippett, 1928; Gnedenko, 1943) encourages asymptotically sample maxima modeling with the GEV distribution." Please tidy up the English here.

Thank you, this part will be rephrased as follows: "Indeed, similarly to the central limit theorem that motivates to model means obtained from different samples using a normal distribution, the Fisher–Tippett–Gnedenko theorem (Fisher and Tippett, 1928; Gnedenko, 1943) encourages to model maxima using the GEV distribution."

RC3#42. l.111: "Y" Realised values of a random variable Y are usually denoted by a lower case letter y_i; l.112 This is meant to denote the rand var indeed.

Thank you for this comment. We will modify this sentence as follows: "if Y is the random variable representing annual maxima, we can assume…".

This notation is sometimes used to indicate that the cdf is defined only for positive values of $1+\xi(y-\mu)/\sigma$, when $\xi$ is not equal to 0 (see, e.g., Eq. 1 in Coles and Perrichi, 2003).

RC3#44. l.116 "Due to these theoretical justifications, the GEV distribution enables a robust estimation of return levels" This is a dogma at best. The variance and bias of estimates depends on the amount of data and the tail probability 1/T associated with the estimated return level.

This comment was also raised by the reviewer RC2 (see comment RC2#7). We agree that we do not provide strong results showing a particular robustness of the return level estimates. We propose to remove it when it appears in the manuscript.

RC3#45. l.119 "for the design working life of building" please make sure the English is correct.

Thank you for this comment, this part of the sentence will be replaced by "for building design".

RC3#46. l.124. "the smoothed anomaly of GMST." to be removed.

Ok, this will be removed.

RC3#47. l.124. "log link function" Provide reference where the link function is defined.

It is defined in Eq. 2 where we indicate that the log transform of the scale parameter is a piecewise linear function of T. To clarify this point, we will move this sentence after the equation.

RC3#48. l.125 "numerical optimization." What does it refer to? I suppose the maximum search for MLE.

Yes, this is correct, we will precise this point with a reference to section 3.3.

RC3#49. l/126. "$T\_t^{obs}$" I don't think you need to write t here.

The subscript $t$ indicates that the global temperature is obtained for the year $t$. It is actually missing at l.121 above and will be added.

RC3#50. l.126. For linear piece l, the slope is sum_i=1^l mu_i, right?

For a linear piece l, the slope is mu_l, and starts from the end of the previous linear piece (l-1).

RC3#51. l.127. "θ is vector of coefficients" What coefficients? Also, the realised values don't depend on anything. Please be careful with the notation.

Thank you for this comment. θ is actually the vector of parameters corresponding to the linear pieces, i.e. {$\mu_i$, $\sigma_i$, $\xi_i$, i=1,...,L}, this will be clarified.

RC3#52. l.129. "$\kappa_2$" You also have kappa_1 in eq. (2).

$\kappa_1$ corresponds to $T_{min}$, and is the starting point of the piecewise linear function. In this sentence, we simply explain where the slope of the piecewise linear function is changing. However, the subscript for $\kappa_{L-1}$ was incorrect and will be replaced by $\kappa_L$.

RC3#53. l.131. "Let" This sentence could be put more concisely.

We agree, this sentence will be rephrased.

RC3#54. l.131. "Maximum" -> maxima.

Here, we refer to a single maximum.

RC3#55. l.131. "pair" you can loose this, the reason why you would introduce k is to denote a pair.

Thank you for this suggestion, this will be done.

RC3#56. l.133-143. "GCM$^{pair\ k}$" k is not a co-variate, it's an ARBITRARY label to identify a model. "aim at adjusting the distribution of GCM-RCM pairs w.r.t. the distribution of the past observations." I cannot see where in eq. (3) the observations are involved. "adjustment coefficient" Up to here i still don't know what is an "adjustment coeff". "assume that these adjustment coefficients are constant, i.e. the same for historical and future climates." I guess this means that you have one set of coeff, not one set per piece of the piece-wise lin fun. However, why not? I didn't read (Brown et al. 2014) to see if they justify this assumption. "$\mu_{GCMi}$" okay, but what is the functional form, i.e., the co-variate dependence. What is the co-variate to start with? Is it the integer i?

This paragraph will be strongly modified in order to clarify how these adjustement coefficients are introduced and to explain the rationale behind this use. The adjustement coefficients are additional parameters that assume that the different members of the ensemble share the same nonstationary GEV model up to a constant shift for the location and scale parameters (see proposed illustration in Fig. 3, in response to the comment RC2#2 above). This shift can be shared by all members ("One for all GCM-RCM pairs"). In this case, there is a single parameter that represents a common shift for all members of the ensemble with respect to the observations. We also consider the possibility of having different shifts for each GCM or for each RCM ("One for each GCM","One for each RCM"), and for all members ("One for each GCM-RCM pair"). Obviously, adding many parameters to our model is only desirable if it leads to an important improvement of our predictive skills, evaluated using the split-sample experiment. As it is not often the case in our application (see Fig. 3 of the manuscript), we did not test more complex forms for the adjustement coefficients. This argument will thus replace the reference to Brown et al., 2014 in the revised manuscript. Again, it must be clear that the adjustement coefficients are directly estimated by maximising the likelihood (4) and **not** by first fitting individual GEV models to each climate simulation.

RC3#57. l.137. "because it sometimes leads to prediction failures." This is a very opaque explanation.

See response to comment RC3#24 above.

RC3#58. l.146 "all annual maxima of a given massif, i.e. annual maxima from the observations and from the 20 GCM-RCM pairs" This is the first time you give indication how you mean to use the obs and model data together.

Otherwise, i'm not at all convinced if this is any good solution to the problem that models are different from each other and from reality. You just throw reality into the mix. But if you have more and more model data, it matters less and less that you have a seed of the truth.

Thank you for this comment which is also related to the comment RC1#2 above. We agree that the combination of the observations and of the different GCM-RCM pairs in Eq. (4) is simplistic and could be improved since the information provided by the observations is diluted by the large size of the ensemble (number of members times the length of the projections). We will discuss these limitations more clearly and in more depth in the revised paper. Different tests have been carried out in order to put more weights on the observations but the formulation of these weights was problematic. However, we see our contribution as a first step in that direction.

RC3#59. l.149. Eq. 4. We really don't need this formula. It's enough to say that you perform MLE for all the data, obs and model. Or, it would be enough, had it been worth to do, as i commented above. Furthermore, you forget to mention that you also do the fitting to individual model pairs too, or various subsets of your data.

We prefer to keep this equation that helps the reader to understand how observed maxima and simulated maxima from the projections are combined (despite the limitations of this approach). It is true that the fitting is also done for various subsets of the data for the evaluation experiments, and this will be added. Nonstationary GEV models are fitted to each climate simulation only for the sake of illustration in Figure 4 and it would be confusing to add this explanation at this stage of the paper in our opinion.

RC3#60. l.153-154. "the calibration of the non-stationary GEV distribution." no idea what this is.

We agree that this part of this sentence was unclear and not necessary, it will be removed.

RC3#61. l.169. Unfortunately, i do not understand what you mean by evaluating predictive performance here.

As indicated in Gneiting and Raftery (2007), the log-score is used to assign "a numerical score based on the predictive distribution and on the event or value that materializes", in our case, the predictive distribution is obtained from the GEV distribution defined in Eq. (2) with an estimated parameter vector $\hat{\theta}$ and the value used to evaluate the predictive performance (the future data of the pseudo-observations in the model-as-truth experiment, and the observations which have been discarded from the dataset fitting in the split-sample experiment). Additional explanations and the reference to Gneiting and Raftery (2007) will be added to the revised manuscript.

RC3#62. l.171. "we select one parameterization" Rather, the selection determines the set of data you will fit.

The final GEV model is always obtained using the observations dataset and the ensemble of climate simulations composed of 20 GCM-RCM pairs. Maybe this was unclear in the original version of the manuscript and we hope that the revised manuscript will clarify this point.

RC3#63. l.171-172. "we select the number of linear pieces with a model-as-truth experiment using zero adjustment coefficients for the GEV parameters" not sure how this would be done

The assumption is that the model-as-truth experiment gives a first indication on the global evolution of the GEV parameters, based on the predictive performance of the long climate runs (1950-2100), while past observations are often limited to assess these evolutions. For this reason, most of the nonstationary GEV models fitted on past observations assume linear trends for the GEV parameters (usually the location and the scale). Our study proposes an approach to assess more complex evolutions of the GEV parameters based on a climate ensemble, even if some limitations must be acknowledged.

RC3#64. l.179. "uncertainty interval" confidence interval it's called. However, for the said reason i'm not optimisitic it's any meaningful .

Thank you for this comment, we will replace "uncertainty interval" by "confidence interval" in the revised manuscript.

RC3#65. l.181. Eq 5. On line 118 you already have this eq. This is a rather unnecessary duplication.

We agree that this equation can appear unnecessary if the reader is familiar with nonstationary GEV models. However, on l.118, the return level is provided for a stationary GEV distribution while the return level depends on *T* in Eq. 5. Furthermore, it seems important to stress that these return levels are not based on the adjustment coefficients.

RC3#66. l.226. Seeing the discontinuity of the slopes for many individual model pairs i wonder if the piece-wise lin model is really good. Perhaps the chi-squared test should really be done. Coles notes what to do in nonstationary EVS, which is what i also followed: https://nhess.copernicus.org/preprints/nhess-2020-117/nhess-2020-117.pdf

Thank you for this comment. First, from the comments above, it seems that there was a misunderstanding about these individual fittings which are used solely for the sake of illustration. We never fit GEV models to individual GCM-RCM pairs in the rest of the paper. The gray curves in Fig. 4 showing the return levels obtained from these individual GEV models are not very smoothed indeed, maybe due to the lack of robustness of these fittings. It is in fact one important motivation for the joint inference of the likelihood (4). The goodness-of-fit of the final GEV models (colored curves in Fig. 4) have been carried out using the Anderson-Darling tests (see comment RC1#1) and these results will be discussed in the revised manuscript. Finally, we agree that an interesting approach for nonstationary GEV models is to express the GEV parameters as a linear combination of additional covariates. This is an approach that we have also considered in other applications (see, e.g. Evin et al., 2021b, for an application to extreme avalanche cycles). When there is a clear relationship between the statistical properties of the extreme variables and some climate descriptors (as is the case in the aforementioned paper between extreme cold temperatures in Europe and the arctic oscillation index), this approach is particularly powerful. However, this is not the case in our application to extreme snow loads in the French Alps where the relationships between climate indices and climate extremes related to precipitation events are often very weak (see Belkhiri and Kim, 2021). A possible idea would be to use the statistical properties of the climate simulations as covariates. These possible avenues will be discussed in the revised version of the manuscript.

RC3#67. l.266. "100 maxima" That does not sound very many.

Our point was that 20 x 100 maxima is a large amount of information compared to the 61 observed maxima.

RC3#68. l.287 "The 90% uncertainty intervals" Why not the usual 95% but a 90% CI?

This 90% CI was preferred to rely on the 5% and 95% quantiles instead of 2.5% and 97.5% quantiles which are estimated with more uncertainties.

References

Belkhiri, L., and T.-J. Kim. 2021. "Individual Influence of Climate Variability Indices on Annual Maximum Precipitation Across the Global Scale." *Water Resources Management* 35 (9): 2987–3003. https://doi.org/10.1007/s11269-021-02882-8.

Coles, S., and L. Pericchi. 2003. "Anticipating Catastrophes through Extreme Value Modelling." *Journal of the Royal Statistical Society: Series C (Applied Statistics)* 52 (4): 405–16. https://doi.org/10.1111/1467-9876.00413.

Drótos, G., T. Bódai, and T. Tél. 2015. "Probabilistic Concepts in a Changing Climate: A Snapshot Attractor Picture." *Journal of Climate* 28 (8): 3275–88. https://doi.org/10.1175/JCLI-D-14-00459.1.

Evin, G., S. Somot, et B. Hingray. 2021a « Balanced Estimate and Uncertainty Assessment of European Climate Change Using the Large EURO-CORDEX Regional Climate Model Ensemble ». *Earth System Dynamics* 12, n° 4: 1543- 69. https://doi.org/10.5194/esd-12-1543-2021.

Evin, G., P. D. Sielenou, N. Eckert, P. Naveau, P. Hagenmuller, and S. Morin. 2021b. "Extreme Avalanche Cycles: Return Levels and Probability Distributions Depending on Snow and Meteorological Conditions." *Weather and Climate Extremes*, July, 100344. https://doi.org/10.1016/j.wace.2021.100344.

Gneiting, T., and A. E Raftery. 2007. "Strictly Proper Scoring Rules, Prediction, and Estimation." *Journal of the American Statistical Association* 102 (477): 359–78. https://doi.org/10.1198/016214506000001437.

Hu, G., T. Bódai, and V. Lucarini. 2019. "Effects of Stochastic Parametrization on Extreme Value Statistics." *Chaos: An Interdisciplinary Journal of Nonlinear Science* 29 (8): 083102. https://doi.org/10.1063/1.5095756.

June-Yi Lee, Tamás Bódai. (2021) Indian summer Monsoon Variability 1st Edition, El Niño-teleconnections and beyond: Chapter 20, Future Changes of the ENSO-Indian Summer Monsoon Teleconnection, Elsevier, pp. 393-412.

Rajczak, J., and C. Schär. « Projections of Future Precipitation Extremes Over Europe: A Multimodel Assessment of Climate Simulations ». *Journal of Geophysical Research: Atmospheres* 122, n° 20 (2017): 10,773-10,800. https://doi.org/10.1002/2017JD027176.

Ribes, A., S. Qasmi, and N. P. Gillett. 2021. "Making Climate Projections Conditional on Historical Observations." *Science Advances* 7 (4): eabc0671. https://doi.org/10.1126/sciadv.abc0671.

Serinaldi, F., and C. G. Kilsby. « Rainfall Extremes: Toward Reconciliation after the Battle of Distributions ». *Water Resources Research* 50, n° 1 (1 janvier 2014): 336- 52. https://doi.org/10.1002/2013WR014211.

Storch, Hans von, and Francis Zwiers. 2013. "Testing Ensembles of Climate Change Scenarios for 'Statistical Significance.'" *Climatic Change* 117 (1): 1–9. https://doi.org/10.1007/s10584-012-0551-0.

Tél, T., Bódai, T., Drótos, G., Haszpra, T., Herein, M., Kaszás, B., Vincze, M. (2020) The Theory of Parallel Climate Realizations – A New Framework of Ensemble Methods in a

Changing Climate: An Overview, Journal of Statistical Physics, http://doi.org/10.1007/s10955-019-02445-7

Verfaillie, D., M. Lafaysse, M. Déqué, N. Eckert, Y. Lejeune, et S. Morin. « Multi-component ensembles of future meteorological and natural snow conditions for 1500m altitude in the Chartreuse  mountain range, Northern French Alps ». *The Cryosphere* 12, n$^o$ 4 (10 avril 2018): 1249- 71. https://doi.org/10.5194/tc-12-1249-2018.

Vionnet, V., E. Brun, S. Morin, A. Boone, S. Faroux, P. Le Moigne, E. Martin, et J.-M. Willemet. « The detailed snowpack scheme Crocus and its implementation in SURFEX v7.2 ». *Geosci. Model Dev.* 5, n$^o$ 3 (24 mai 2012): 773- 91. https://doi.org/10.5194/gmd-5-773-2012.

---

## Author Response (AR1)

**Point by point reply to the Interactive comments of the Referees**

Dear Editors,

We thank the referees for their thorough reviews and for the numerous suggestions that helped us to improve the manuscript.

Please find below, point by point, our answers to all the other suggestions. Referee #1 suggestions are in red, suggestions from Referee #2 are in blue, while suggestions from Referee #3 are in green.

We hope that our revised manuscript will be found suitable for publication in "Earth System Dynamics"

Yours sincerely,

On behalf of the co-authors

Guillaume Evin

**Comments from Referee #1**

The paper raises a very important issue: in order to obtain robust estimates, especially in a nonstationary setting, we need to use a multitude of available information sources. However, it is a challenge to connect properly these different information sources and requires a lot of scientific effort. I consider this paper to be one of the initial steps in the right direction regarding the quantification of changing extreme events due to global warming. The paper has a clear structure, the language is understandable despite the many technical details and steps, the figures are illustrative. I appreciate a lot that the authors discuss critically the drawbacks of their method in Sec. 5.2, and that they provide a clear and concise overview of related studies and how this work is embedded in the current scientific literature. Another strength of this work is that a range of possible parametrisations is explored and the optimal one is chosen in an objective way, based on a mean logarithmic score. I tend to suggest the manuscript to be accepted for publication, but only after a few very important issues are clarified.

We thank the reviewer #1 for this positive feedback and these useful comments on our manuscript. Please find below a detailed response to individual comments and questions.

Main comments:

RC1#1. My main criticism is that the authors use a non-stationary GEV setting without showing that the annual maxima of snow load are properly estimated by a GEV distribution. The authors write that "...due to these theoretical justifications, the GEV distribution enables a robust estimation of return levels". Yes, the GEV distribution enables (under certain conditions) a robust estimation, but only if:

1) the chosen block size $n$ is large enough, as the theory applies for $n \to \infty$. The block size of one year is not automatically large enough (one might need 2 years or 10 years or even longer)–it has to be shown that the annual maxima can be reliably estimated based on a GEV distribution (see convergence plots and diagnostic plots in Coles, 2001).
2) the auto-correlation is weak enough -the stronger the auto-correlation, the larger the smallest block size for which the GEV limit is valid. These convergence issues are discussed extensively in Galfi et al.(2017, Complexity).

I ask the authors to show that the convergence to the GEV distribution is good enough for the chosen block size, i.e. that the annual maxima are properly modelled by a GEV distribution. This should be done before building up the non-stationary framework. If this cannot be shown, I'm afraid that the whole experimental setup described in the paper is useless, as it is a necessary condition. In the theoretical description of the GEV distribution in Section 3.1 more emphasize is needed to underline the asymptotic nature of the theory.

We agree with the reviewer that we make the strong assumption that annual maxima follows a non-stationary GEV distribution whereas our block size is limited to one year. As done in many studies on climate extremes, the choice of annual blocks represents a compromise between having sufficiently large blocks in order to obtain maxima on a large number of values (365 days), and not too large in order to have enough blocks to avoid very uncertainty GEV parameter estimates (which would be the case with a 10-year block size, leading to very small samples). The revised version of the manuscript clarifies this pragmatic choice as follows:

*"In theory, the GEV distribution is adequate when maxima are computed over blocks of infinite size. In practice, the GEV distribution is usually applied to annual maximal and has been shown to provide reliable estimates of return levels in many hydrometeorological*

*applications (Coles, 2001; Katz et al., 2002; Cooley, 2012; Papalexiou and Koutsoyiannis, 2013)."*

We recognize that we may not totally fulfill the convergence conditions, but that, according to the data, our model choice appears to be sensible. The revised manuscript now presents a quantitative evaluation of the goodness of fit. We rely on the Anderson–Darling statistical test (see Appendix A), which is the most powerful test for the Gumbel distribution (Abidin et al., 2012), similarly to what was proposed in Le Roux et al. (2021). This test assesses if the residuals follow a standard Gumbel distribution. For every selected model, the p-value of this test was computed for each GCM-RCM pair separately (and the observations) for each massif. In the Figure below, we observe that the test is rejected for 20% of the 483 cases (23 massifs x 21 time series). This relatively high number seems to be mainly explained by the small values reached at the end of the century for many GCM-RCM runs. Indeed, the same tests applied at an elevation of 2700 m (see Figure 2 below) show a much smaller percentage of rejections (7%) and larger p-values. These results are provided in section 5.2 of the manuscript and the inadequacy of the selected GEV models to represent these small values for some GCM/RCM runs is discussed:

*"The inadequacy of the selected GEV models to represent these small values can be related to the high proportion of zero snow load values at the end of the century. In these cases, as annual maxima represent maxima of a limited number of positive values, the asymptotic nature of the extreme value theory might be not respected. One alternative could be to consider larger block sizes (maxima over several years). However, the smaller sample sizes would lead to more uncertain parameter estimates. The application of extreme value models to bounded variables thus remains an important challenge, especially in a context of climate change where this issue only affects a small part of the dataset."*

The fact that snow loads maxima reach a lower bound at the end of the century is clearly a major challenge for this application, and is related to the difficulties in adjusting the shape parameter (see Section 4.2). However, we emphasize the fact that this application can be considered as a non-trivial example of the proposed methodological approach, which also illustrates its limitations. An application to unbounded variables (e.g. annual maxima of temperatures) would be easier to present but would be less innovative in terms of specific application, and would fail to illustrate these challenges.

All in all, we firmly believe that these results are sufficient to support our modeling choices with regards to our application, and to illustrate the potential of the framework we propose for many applications.

[Figure]

Figure 1: Distribution of p-values for the Anderson-Darling test for the elevation 1500 m.
For every selected model, a p-value was computed for each GCM-RCM pair separately.

[Figure]

Figure 2: Distribution of p-values for the Anderson-Darling test for the elevation 2700 m. For every selected model, a p-value was computed for each GCM-RCM pair separately.

RC1#2. It is a bit disappointing that even after performing the adjustments and applying the complex methodology, the results still do not follow the observations, as shown by Fig. 4 and discussed shortly in the text. One reason for this could be that, as the authors explain, the evaluation set for adjustment parametrization contains only a few (24, 17,10) maxima, thus the selection of the optimal parametrisation might be misleading. I believe a more thorough discussion is needed here. It is also not clear for me how to overcome the issue regarding the estimation of the shape parameter: in case of no adjustment, the model results do not follow the observations, in case it is adjusted it can lead to prediction failures. The authors do not suggest any solution for this, although it would be crucial for the applicability of the method in future studies.

It is true that this issue is particularly challenging and open to discussion. On the one hand, we can imagine some applications where the model should fit closely the observations, for example if a long series of past observations is available or if it is assumed that only past observations provide a relevant information about the tail of the distributions (i.e. if it is assumed that climate simulations are not able to simulate reliable climate extremes). A simple solution for doing this could be to put more weight on the observations in the likelihood (Eq. 4) in comparison to the climate simulations. However, this approach has not been tested thoroughly. More research is needed to provide a convincing solution for this issue, as now acknowledged in the last paragraph of the conclusion:

"*Finally, further work is needed to obtain a better agreement between the non-stationary GEV model representing the ensemble of maxima from climate projections and past observed maxima. Indeed, observed maxima are mainly used to identify and correct strong*

*disagreements between the observed and simulated maxima, using adjustment coefficients, and the fitted non-stationary GEV model is not really constrained to represent observed maxima. Bayesian approach representing the predictive distribution of climate projections conditional on historical observations (Ribes et al., 2021) could be a possible approach to better constrain a GEV model. However, it would require i) long series of past observations to identify the relationships between the forced climate responses to greenhouse gases obtained from the climate models and from the observations. ii) an adaptation of the Gaussian framework in order to comply with the distribution of extreme values (i.e. GEV distributions).*"

RC1#3. It is not totally clear for me why is it important to know return levels of a variable whose extremes are expected to decrease with global warming (assuming a simple direct relationship between them), thus becoming less extreme? I think the authors should put more emphasize to show the relevance of this subject.

Thanks for this question. Beyond the methodological contribution of our submission, our second objective is to check whether return levels are expected to increase or decrease. Indeed, the literature points to a decrease of mean winter SWE (IPCC 2019), i.e. to a decrease of mean winter snow load. However, to the best of our knowledge, projected trends in extreme snow loads have never been studied before.

Since annual means of snow loads are expected to decrease, our first hypothesis was that extreme snow load would also decrease. However, in cold regions (high elevation regions for instance) we expect extreme snowfall to increase with climate change (O'Gorman 2014), thus our second hypothesis was that this increase of extreme snowfall can lead to an increase of extreme snow load. According to our results, the former hypothesis is the most likely hypothesis in the French Alps.

Even if extremes are expected to decrease, a quantification of these decreases are of prime interest:

- to study compounds extremes, e.g. extreme snow load combined with extreme wind,
- to adapt structures standards, e.g. to decrease the constraints used to design new structures, which may reduce the construction cost.

These motivations have been added to the manuscript at Section 2 "Data".

Specific comments:

RC1#4. L93 Are the time periods "historical" 1951-2005 and "future" 2006-2100 correct?

Yes, these are the standard historical and future time periods used in the EUROCORDEX experiment obtained from a CMIP5 ensemble, the RCP scenarios prescribing greenhouse gas concentration trajectories from 2006.

RC1#5. Eq.4 There should be two equations here: $\theta$=argmax($p(y|\theta)$)and $p(y|\theta)$=∏...

Thank you for this remark, this was a mistake. This equation has been modified.

RC1#6. L177-L178 It is not totally clear what "RL50" stands for because it is written that "is computed without adjustment coefficients", but later in Figure 4 it is used also for the case with adjustment coefficients.

Thanks for this remark. Whether or not the model has adjustment coefficients, RL50 corresponds to the 50-year return level computed without adding the adjustment coefficients.

To clarify that, this paragraph "*In other words, if the selected parameterization has adjustment coefficients, RL50 is computed without these adjustment coefficients since using these coefficients would provide the 50-year return level of the GCM-RCM pairs.*" has been replaced by "*In other words, if the selected parameterization has adjustment coefficients, we do not add these coefficients to compute the RL50.*"

RC1#7. Figure 4: I think that in case of the line with the "warm" colours the legend should be different for the 4 subplots –the legend is the same for each subplot, although the text and the figure caption suggest the opposite.

The legend is not exactly the same for each subplot. Indeed, at the end of the line with the "warm" colors it is either written 'all GCM-RCM pairs', 'each GCM', 'each RCM', or 'each GCM-RCM pair'.

RC1#8. Figure 5: To which year or period do these return levels refer?

As was indicated in the legend of Figure 5, these return levels correspond respectively to the 50-year return levels for T=+1, T=+2, T=+3 and T=+4 degrees of global warming. This figure (now numbered Figure 6) has been modified to indicate more clearly these warming levels. We have also modified Figure 7 to highlight the corresponding changes of global warming in each subplot.

References

Abidin, N. Z., Adam, M. B., & Midi, H. (2012). The Goodness-of-fit Test for Gumbel Distribution: A Comparative Study. Matematika, 28(1), 35–48. Retrieved from https://doi.org/10.11113/matematika.v28.n.313.

Le Roux, E., G. Evin, N. Eckert, J. Blanchet, & S. Morin. (2021) Elevation-Dependent Trends in Extreme Snowfall in the French Alps from 1959 to 2019. The Cryosphere 15, n° 9: 4335‑56. https://doi.org/10.5194/tc-15-4335-2021.

We thank the reviewer #2 for these comments on our manuscript. Please find below a detailed feedback to individual comments and questions.

Main comments:

RC2#1. The methodological approach proposed in the paper, based on simultaneous use of observations and model projections, is stimulating in particular when facing problems (like the one addressed in the paper) in which numerical simulation models are characterized by heavy tuning-parameterization (fudge factors) changing in time. However, in order to make the estimation process tractable in the specific application considered in the paper, many ad hoc assumptions have to be introduced and the results are admittedly (Section 5.2) problematic, raising doubts concerning the **applicability** of the adopted working hypotheses. This situation often occurs in operational statistical estimation: thorough a posteriori analysis is almost invariably required; the authors should critically re-examine their assumptions.

We thank the reviewer for this comment. We agree that some assumptions are particularly critical and Section 5.2 tries to discuss these limitations. Our ambition for this study is to present different possible nonstationary extreme value models adapted to an ensemble of simulations from climate models. It follows a study by the same authors on past observations (Le Roux et al., 2020) which shows that nonstationary GEV models are adequate for the assessment of ground snow load extremes in the French Alps. We also refer to the comment RC1#1 which presents Goodness-of-Fit tests of the proposed models of this study. It is also important to note that the set of the proposed models is larger that most nonstationary GEV models of the literature. These additional ad-hoc assumptions can actually be viewed as a more general framework. For example, the GEV parameters of nonstationary models often follow linear trends, which is a special case of the piecewise linear functions, i.e. when L=1.

Among the wide range of possible nonstationary GEV models, we have performed many tests in order to select a limited set of applicable models. Some models were overparameterized and could not be fitted (i.e. the parameters could not be identified). On the contrary, it must also be pointed out that modeling the evolution of the GEV parameters using nonlinear functions is particularly challenging, which explains that it is rarely done (see Table 1), whereas monotonic evolutions seem overly simplistic in many applications, especially over a long time period (e.g. more than 150 years). As a consequence, we do not wish to re-examine this particular assumption as it both represents an interesting challenge (how to cope with these nonlinear evolutions) and a necessary framework (for some extreme variables that could reach a "peak" in the future).

We also applied the methodology to other variables (results not shown) and obtained satisfying results in the sense that their interpretations were in agreement with what can be expected from a physical point of view. We are thus confident that the general approach (i.e. the two-step approach presented in Section 3.5 and the use of adjustment coefficients) is promising and can be considered as a solid basis for various applications, although we acknowledge that the current approach has some limitations. These limitations are now discussed more thoroughly in Section 5.2 "Methodological choices, assumptions and limitations" and 6 "Conclusions and outlooks".

RC2#2. The paper is neat and clean, but here and there not easily legible as it is very concise ("dense"): since the paper proposes issues of potential interest for a wide audience in which "non experienced" readers could find elements of interest I suggest a more

"friendly" communication approach; but I leave to the authors deciding whether being concise is more important than being readily accessible for a wider audience.

It is important for us to reach a wide audience of practitioners and we agree that the current manuscript could be improved from this perspective. We therefore added the following sketch in order to illustrate the concept of the "adjustment coefficients" in Section 3.2:

[Figure]

Figure 3: Illustration of the evolution of the location parameter μ(T) for the different options of adjustment coefficients with a fictive ensemble composed of 4 different GCM/RCM pairs with 2 different GCMs and 2 different RCMs. (a) location parameter μ(T) if the GEV model was fitted individually to each trajectory. (b) location parameter μ(T)+μ$_k$ using adjustment coefficients.

Specific comments:

RC2#3. Many acronyms and "technical slang" words appear in the paper: a glossary may help.

Thank you for this comment. We define these acronyms when they first appear in the text. If the reviewer thinks that some acronyms or technical terms are not properly defined, we are pleased to explain them in more detail. Some terms have also been removed from the manuscript (e.g. "chains", see the following comment RC2#4).

**RC2#4. Line 3** "chain of MME": define in text or in glossary.

MME refers to "multi-model ensembles" as defined in the text at l. 28 and is a standard acronym when a set of simulations obtained from different climate models is used. It is a very common acronym and it does not seem necessary to define it in a glossary in our opinion. The term "chain" has been replaced by "climate simulation". Climate simulations are often referred to as "chains" because it is obtained as the result of a simulation chain: a RCP scenario provides the greenhouse gas concentration trajectories, then a General Circulation Model (GCM) provides large scale climate simulations for these scenarios, then a Regional Climate Model provides climate simulations at a fine spatial resolution forced by the GCM outputs.

**RC2#5. Line 7** "with a robust quantification of uncertainties.": this claim appears repeatedly in the paper; I found mathematical definition in Appendix A: Uncertainty estimation a technical quantification uncertainties, but not an analysis-discussion of the "robustness" of the estimation itself.

We agree that the term "robust" can be used only if there is a clear statistical interpretation. In our case, the quantification of uncertainties assesses in-sample variability using bootstrap methods and we do not provide results about the robustness of these uncertainties. The term "robust" has been removed in the revised version of the manuscript.

**RC2#6. Line 11** "is of major interest for the structural design of roofs": not only (skying, avalanches, mobility, etc.); a few more words about applications could help.

We agree that snow loads may cause large-scale environmental damages (e.g. to forests) as well as damages to infrastructures (transportation networks, electricity networks) and accidents to people. This has been added at the end of Section 1 "Introduction". This comment about specific applications has been removed from the abstract since this is not a key aspect of the paper.

**RC2#7. Line 24** "EVT makes it possible to robustly estimate return levels": see Line 7 comment.

As the term "robust" is not supported by the results, it has been removed from the manuscript.

**RC2#8. Line 29** "estimated separately on each chain of the MME": see Line 3 comment.

"Chain" has been replaced by "climate simulation".

**RC2#9. Line 32** "30-year time slices": perhaps it is worth mentioning that 30 years is the traditional (WMO) "time scale" of "climatological" analysis.

Thanks for this comment, this has been added.

**RC2#10. Line 52** "robustly quantify uncertainties": see above lines 7 and 24 comments.

Thank you, see responses above.

**RC2#11. Line 63** "adjustment coefficients": a few more words could help.

Yes, we agree, the following paragraph has been added:

*"To this end, we introduce parameters (so-called adjustment coefficients) for the location and scale parameters of the GEV distribution that can account for systematic shifts between the different climate trajectories.*

*Different parameterizations of these adjustment coefficients are tested in order to describe the variability between the climate trajectories, the best parameterization being selected using split-sample tests on past observations."*

RC2#12. **Line 80** "snow load" see Line 11 comment.

See our response to the comment RC2#6.

RC2#13. **Line 92** "Quantile mapping method ADAMONT": a few words about it?

ADAMONT is a bias-correction and downscaling method which aims at adjusting daily climate projections from a regional climate model against a regional reanalysis of hourly meteorological conditions using quantile mapping. These explanations have been added to the manuscript.

RC2#14. **Line 204** "For a detailed analysis of the mean logarithmic scores of each parameterization for each massif, see Supplement, Part C.": what is Supplement, Part C? Where is it?

Thank you for pointing out this mistake. This supplement was part of a previous version of the manuscript. This sentence has been removed.

RC2#15. **Fig.4** This figure plays a central role in the paper: some graphical features are too faint.

Thank you for this relevant comment. It is true that the uncertainty bands were too faint and the figure has been modified in order to improve these graphical features. In particular, thick lines have been added to clearly show the contours of these uncertainty bands.

RC2#16. **Line 255** "Figure 2.3 of IPCC (2019)": wouldn't it be possible to insert this figure or its direct internet link in the text?

Thank you for this suggestion. The following permalink has been inserted in the revised manuscript:

https://www.ipcc.ch/srocc/chapter/chapter-2/2-1introduction/ipcc-srocc-ch_2_3/

RC2#17. **Line 273** "because it sometimes leads to prediction failure, i.e. the predictive distribution gives a null probability to some future annual maxima.": this is not clear to me!

This situation happens when $\xi(T)<0$, which means that the predictive distribution has an upper bound, and when some future annual maxima lies above this upper bound. This explanation has been added to the text.

RC2#18. **Line 287** "The 90% uncertainty intervals of return levels (Fig. 4) account both for the sampling uncertainty (Appendix A) and the climate model uncertainty (distributions are fitted together from the past observations and all GCM-RCM pairs).": not easy to distinguish in the figure (see comment to Fig.4 above).

We agree that this figure could be improved in order to highlight the differences between the different uncertainty bands. This figure has been modified (see comment RC2#15 above).

References

Le Roux, E., G. Evin, N. Eckert, J. Blanchet, and S. Morin. « Non-Stationary Extreme Value Analysis of Ground Snow Loads in the French Alps: A Comparison with Building Standards ». *Natural Hazards and Earth System Sciences* 20, nᵒ 11 (6 novembre 2020): 2961‑77. https://doi.org/10.5194/nhess-20-2961-2020.

We thank Reviewer #3 for the numerous comments and the in-depth review of our manuscript. We appreciate this expertise although we think that some aspects of our work have been misunderstood, probably due to a lack of (or unclear) explanations in the original version of the manuscript. We firmly believe that the revised version presents more accurately the proposed method, the motivations and its limitations. Please find below our responses to the different comments.

Main comments:

RC3#1. My feeling is that this paper is not providing a good solution to a real problem. They consider several GCM-RCM model pairs ('model' in the following). In the Introduction the authors point out that previous studies evaluated extreme value statistics (EVS) for individual models and in some cases they took then the ensemble mean of return levels. In this regard the authors are concerned that the estimates for separate models are not so reliable because of data scarcity. However, they seem to do this themselves.

Thank you for this comment. One of our main motivations for the proposed method is to improve the estimates using a joint estimation approach. Indeed, fitting individual GEV to each climate simulation leads to a set of very uncertain estimates. Taking the mean (or the median, e.g. Rajczak and Schär, 2017) of very uncertain estimates of a return level is not equivalent to having one estimate of the mean return level obtained from the whole set of climate simulations, as is done in this work. This has been clarified in the revised version of the manuscript after the description of Figure 5 (previously Figure 4) which was perhaps misleading:

"*In addition, we also perform individual fitting to each GCM/RCM pair, the corresponding return levels being shown with thin gray lines. Note that these estimates are shown for illustrative purposes only, and do not contribute to the final return level estimates indicated with warm colors.*"

RC3#2. They introduce the concept of "adjustment coefficients", which is really just a difference of an estimate of the GEV parameter for a particular model (or subset of the data) from the estimate upon lumping all the data. I think we really don't need a name for this, beside the problem that they do what they criticised.

It must be clear that these "adjustment coefficients" are not direct differences between estimates of GEV parameters. They are estimated by maximizing the likelihood function (4) and represent a compromise between fitting past observations and future maxima as obtained from the different climate models. The name for these parameters can of course be debated but they represent the adjustment of a systematic shift between maxima obtained by the different climate models (see Figure 3 above in response to the comment RC2#2). According to the properties of the climate models, this shift can be assumed to be the same for each GCM, or each RCM. We also test the possibility of estimating these parameters for all GCM/RCM pairs. In this application, these parameters do not often lead to the best model, as shown in Figure 4c of the revised manuscript. However, we believe that this possibility can be interesting in other applications where annual maxima from climate models largely differ from observed maxima in the historical period. As indicated in our response to comment RC2#2, a simple sketch of the concept around these parameters has been added to the revised version.

RC3#3. On the other hand, lumping all the data together, in order to have seemingly more robust estimates, is also problematic. As i pointed out in some recent publications of mine, a model ensemble (or multi-model ensemble, MME, as it's most often called) does not

represent an objective probability distribution. As such, fitting a GEV to MME data is flawed methodology. It has no meaningful probabilistic interpretation. On the contrary, doing this for (converged) initial condition ensembles is fine. It is actually a profound scientific challenge on how to use MME data in a meaningful way. I don't mean to discourage anyone from trying, though, and hope that real progress can be made. The authors promised a constraining of the estimates/projections using observational data. Emergent constraints is now a popular concept, but it appears to me that the authors did nothing like that. They simply threw the observational data in the mix. However, the information from it is diluted by the large amount of model data.

Thank you for this comment. We entirely agree that fitting probabilistic models to samples obtained from a multi-model ensemble is questionable, as it is questionable to apply a statistical test to assess the statistical significance of projected averages obtained from a MME (see Storch and Zwiers, 2013). While we understand the advantage of using "Single Model Initial condition Large Ensembles" (SMILEs), MMEs are widely used to assess the impact of climate change because it seems difficult to trust more one particular model (or one GCM-RCM-impact model chain) than the others. Emerging constraints (e.g. Ribes et al., 2021) try to answer this challenge but we agree that the limitations of our approach must be clearly acknowledged. Section 5.2 "Methodological choices, assumptions and limitations" and section 6 "Conclusions and outlooks" have been completed to present these limitations in more depth.

In this work, it is also true that we put the same weight on the observed maxima and the maxima from the climate simulations. As a consequence, it is true that observed maxima have a limited influence on the estimated parameters. Some attempts to put more weights on the observations did not lead to satisfying results because of the limited period covered by the observations, which does not provide much information about the evolution of the GEV parameters. This limitation is also discussed in our response to comment RC1#2 where we indicate that the following paragraph has been added to the revised manuscript:

"*Finally, further work is needed to obtain a better agreement between the non-stationary GEV model representing the ensemble of maxima from climate projections and past observed maxima. Indeed, observed maxima are mainly used to identify and correct strong disagreements between the observed and simulated maxima, using adjustment coefficients, and the fitted non-stationary GEV model is not really constrained to represent observed maxima. Bayesian approach representing the predictive distribution of climate projections conditional on historical observations (Ribes et al., 2021) could be a possible approach to better constrain a GEV model. However, it would require 1. long series of past observations to identify the relationships between the forced climate responses to greenhouse gases obtained from the climate models and from the observations. 2. an adaptation of the Gaussian framework in order to comply with the distribution of extreme values (i.e. GEV distributions).*"

RC3#4. Obs data is rather used for bias correction. I'm not sure this was done. Or, if it was done, then it seems to have even less use to throw the obs data into the mix for doing EVS.

A quantile mapping method (ADAMONT) is a bias-correct method that was applied to climate simulations. This method shows some limitations to correct the tail of the distribution since, by definition, extreme values are rarely observed. This explains why some biases can still be expected in the bias-corrected climate simulations.

RC3#5. I share my detailed comments on the manuscript with the authors in an annotated pdf. Hopefully it is useful one way or another. I'm sorry that i cannot be more positive this time. If i misunderstand something, i'm happy to learn from the author's response.

We appreciate these comments and are also happy to discuss them and defend our position.

Specific comments in the annotated pdf

RC3#6. l.3: chain of the MME?

As indicated in our response to comment RC2#4, this term is sometimes used to refer to a particular climate simulation chain (e.g. scenario/GCM/RCM). This term has been removed from the revised manuscript.

RC3#7. l.7: flexible?

"Flexible" was used in the sense that the proposed nonstationary GEV model can be adapted to a great variety of nonlinear evolution thanks to the piecewise functions. This term has been removed since it does not seem to be informative.

RC3#8. l.8 "functions of time"?

As indicated in section 3.2, the GEV parameters are functions of anomalies of global mean temperature. This has been clarified.

RC3#9. l.8. "adjustment coefficients"?

These adjustment coefficients are additional parameters (see our response to the comment RC3#2).

RC3#10. l.17. Maybe this should go in the end of the sentence.

Thank you for this suggestion, this has been done.

RC3#11. l.19: I cannot think anything else.

"One of" has been removed from the revised version.

RC3#12. l.19: Even the most basic climatic means are obtained/projected by simulations.

We agree with the reviewer.

RC3#13. l.23: I would say that quantiles and tail probabilities are more often used by climate scientists than EVT.

It is true that return levels are often obtained using different empirical formulas (e.g. Gringorten) or various distributions. EVT seems to be dominant in the past ten years (see, e.g., Serinaldi and Kilsby, 2014, concerning rainfall extremes). We have replaced "usually" by "often" to indicate that the EVT approach is undoubtedly popular for the assessment of climate extremes.

RC3#14. l.24. Why robustly? If we are interested in too high return levels, surely the estimates can be very poor.

This comment was also made by reviewer #2 (see comment RC2#7) and we agree that "robustly" was not adequate, it has been removed.

RC3#15. l.29. I don't know what you mean by chain. I guess you mean time series. If so, "chain" is not used in English in this context.

The term "chain" has been removed (see also our response to the comment RC2#4).

RC3#16. l.32. 30 years is not really a slice, is it?

"slices" has been replaced by "periods".

RC3#17. l.44 Do we really need this reference for nonstat EVT? This seems to unnecessarily inflate the bibliography. There is Coles already cited.

This reference has been removed.

RC3#18. l.47 chain

This has been replaced by "climate simulation".

RC3#19. l.51 This is poor English. Otherwise, this concept is unfortunately flawed. We can only fit a probability distribution if we have reason to believe that there is an objective physical measure from which our data is drawn from. However, regarding a MME there isn't. It is an ensemble of opportunity (or chance). There isn't a probability space from which Earth System Models are samples. This is what we were articulating in another context of forced response. The right approach is considering Initial condition large ensembles because in that case there is an objective physical probability measure generated by the one climate model that represents some hypothetical physics.

This part of the sentence has been rephrased and replaced by "on a temporal non-stationary GEV model fitted to all ensemble members". See our response to comment RC3#3 concerning the rest of this remark.

RC3#20. l.53 why this reference?

This reference has been removed.

RC3#21. l.56 This should not be assumed but rather checked. What one would check is whether the ensemble still depends on its initialisation or already converged to the snapshot attractor. The check can proceed by checking the stat significance of the distributions (concerning a particular scalar observable) bw TWO ensembles that were initialised differently. (Drotos et al. J Clim 2015).

We agree but it does not seem that convergence has been checked in the papers cited in Table 1 (Kharin and Zwiers (2004), Wang et al. (2004), Aalbers et al. (2018), Fix et al. (2018)).

RC3#22. l.57. On l 51-52 the text reads such that you follow this approach.

Thank you for this comment. This sentence was misleading and has been modified.

RC3#23. l.60 How does this solve the described problem?

Using past observations and adjustment coefficients, this non-stationary GEV model assumes that different climate simulations can be used to project future return levels. This was perhaps unclear from this version of the manuscript and we hope that the proposed amendments will motivate this approach (see also the response to comment RC3#2).

RC3#24. Why do you exclude the shape parameter from this treatment?

This is explained later at l. 137: "*because it sometimes leads to prediction failures.*" for example when a negative shape parameter leads to predictive GEV distribution with an upper bound and when some maxima exceeds this upper bound. This explanation has been added.

RC3#25. l.63. By this do you mean "the various GEV distributions of all GCM-RCM pairs"? Variability of climate trajectories does not sound anything like that.

Here, we mean that a MME obtained with many different climate models can lead to a large variation of projected changes. This is true for average changes (see, e.g. Evin et al., 2021a for a EURO-CORDEX RCM ensemble) and for extremes (Rajczak and Schär, 2017).

This refers to the joint estimation over all simulations of the MME but "jointly" was not really needed here and has been removed.

RC3#27. l.65 Why do you say that past observations represent the most likely climate trajectory? This is so strange language to me. And i'm not convinced that it makes deep sense.

Thanks for this comment. We agree that "*to represent the most likely climate trajectory*" was not clear, it has been removed.

RC3#28. l.65 By this do you mean that you fit output data from individual models whereby you have different e.g. location parameter estimates for the different models? If so, there seems to be no difference from what you describe on l46. Rather, you stop short of proceeding with obtaining some ensemble statistics, e.g. mean of the return levels.

Thank you for this comment. As indicated above, the purpose of these adjustment coefficients is to account for systematic shifts in climate simulations, either for the location or the scale parameter. However, we consider different options for these coefficients (one by GCM, one by RCM, etc.) and this is now illustrated in Figure 3 with a simple sketch. However, we insist on the fact that these adjustment coefficients are part of the joint inference and are **not** obtained by first fitting individual GEV models to the different climate simulations.

RC3#29. l.68. Does the data look piecewise linear? If not, why not using some polynomial?

Polynomial functions have been tested but can lead to poor models for some types of evolution. In particular, they are not very well suited to changes of trends (e.g. an increase followed by a decrease). Piecewise linear functions provide a certain flexibility in this regard.

RC3#30. l.76 What does this mean. I suppose you might need to provide a reference here.

This is explained at l.93-94. "Statistically adjusted" has been removed from this sentence.

Thank you for this comment, this has been corrected.

RC3#32. It would be good to have a short explanation what it is so that one doesn't need to look it up.

We have added the following definition "the pressure exerted by the snow".

RC3#33. l.85. It would be helpful to say that snow load has the physical unit of pressure, that is Pascal.

Snow load usually has the units of kN/m2. This has been added.

RC3#34. l.85. Otherwise, one has to wonder if this study is very useful in case it is very common to have different snow depths just meters apart, say, from house to house or street to street.

This is a relevant comment and it is true that snow depth is known to vary a lot in space. However, aside from these measurement issues, there is a clear need to assess snow load hazards as it is indicated in the introduction (e.g. for the design of roofs) and in our response to the comment RC2#6, see also, e.g. Croce et al., 2018. In our study, snow load data are actually obtained using a reanalysis method which has been evaluated against independent in-situ snow depth observations (see the recent study by Vernay et al., 2021, https://essd.copernicus.org/preprints/essd-2021-249/, recently accepted for publication)

RC3#35. l.85. If i wanted to see what a particular model has in a particular model, i could not find it. Perhaps do not connect the lines.

We agree that it is difficult to look at a particular climate simulation due to the many trajectories. Here the figure mainly aims at showing the overall variability from one year to the other and between the different members of the ensemble.

RC3#36. l.92. "The" is missing.

"The" has been added.

RC3#37. l.92. software package Crocus?

Thank you for this comment. Crocus is actually the part of the S2M numerical chain that produces the evolution of the snowpack. This precision has been added to the manuscript.

RC3#38. l.99. "The" missing".

This has been added.

RC3#39. l.99. "Observational"?

Thanks, this has been corrected.

RC3#39. l.101. Is it because this way you have a much better "stat agreement" bw. the models?

It is true that there is a better agreement between the statistical properties of the snow load maxima produced by the different climate models when the covariate is the global temperature than when it is time. Indeed, the climate sensitivity of each GCM can produce large differences of warming for a particular time horizon (e.g. 2050). Taking a warming horizon (e.g. +2°C) removes this "climate sensitivity effect" and makes the different projections more consistent (see also Verfaillie et al., 2018).

RC3#40. l.103. "does not depend on the internal variability of GMST" I don't understand what you mean.

We mean that the "smoothed GMST" is not affected by the internal variability simulated by the GCM (see the fluctuations of the raw GMST at the decadal scale in Fig. 2).

RC3#41. l.109-111: "Indeed theoretically, as the central limit theorem motivates asymptotically sample means modeling with the normal distribution, the Fisher–Tippett–Gnedenko theorem (Fisher and Tippett, 1928; Gnedenko, 1943) encourages asymptotically sample maxima modeling with the GEV distribution." Please tidy up the English here.

Thank you, this part has been rephrased as follows: "*Indeed, similarly to the central limit theorem that motivates to model means obtained from different samples using a normal distribution, the Fisher–Tippett–Gnedenko theorem (Fisher and Tippett, 1928; Gnedenko, 1943) encourages to model maxima using the GEV distribution.*"

RC3#42. l.111: "Y" Realised values of a random variable Y are usually denoted by a lower case letter $y\_i$; l.112 This is meant to denote the rand var indeed.

Thank you for this comment. We have modified this sentence as follows: "*if Y is the random variable representing annual maxima, we can assume…*".

RC3#43. "u+ denotes max(u,0)" Where is this coming from?

This notation is sometimes used to indicate that the cdf is defined only for positive values of $1+\xi(y-\mu)/\sigma$, when $\xi$ is not equal to 0 (see, e.g., Eq. 1 in Coles and Perrichi, 2003).

RC3#44. l.116 "Due to these theoretical justifications, the GEV distribution enables a robust estimation of return levels" This is a dogma at best. The variance and bias of estimates depends on the amount of data and the tail probability 1/T associated with the estimated return level.

This comment was also raised by the reviewer RC2 (see comment RC2#7). We agree that we do not provide strong results showing a particular robustness of the return level estimates. We have removed this term from the manuscript.

RC3#45. l.119 "for the design working life of building" please make sure the English is correct.

Thank you for this comment, this part of the sentence has been replaced by "*for building design*".

RC3#46. l.124. "the smoothed anomaly of GMST." to be removed.

Ok, this has been removed.

RC3#47. l.124. "log link function" Provide reference where the link function is defined.

It is defined in Eq. 2 where we indicate that the log transform of the scale parameter is a piecewise linear function of T. To clarify this point, we have moved this sentence after the equation.

RC3#48. l.125 "numerical optimization." What does it refer to? I suppose the maximum search for MLE.

Yes, this is correct, we now precise this point with a reference to section 3.3.

RC3#49. l/126. "$T\_t^{obs}$" I don't think you need to write t here.

The subscript *t* indicates that the global temperature is obtained for the year *t*. It is actually missing at l.121 above and has been added.

RC3#50. l.126. For linear piece l, the slope is sum_i=1^l mu_i, right?

For a linear piece *l*, the slope is *mu_l*, and starts from the end of the previous linear piece (*l*-1).

RC3#51. l.127. "θ is vector of coefficients" What coefficients? Also, the realised values don't depend on anything. Please be careful with the notation.

Thank you for this comment. θ is actually the vector of parameters corresponding to the linear pieces, i.e. {$\mu_i$, $\sigma_i$, $\xi_i$, i=1,...,L}. We agree that the previous notations were unclear as θ was also used to denote the entire set of parameters, with the adjustment coefficients. The entire set is now denoted by Θ.

RC3#52. l.129. "$\kappa_2$" You also have kappa_1 in eq. (2).

$\kappa_1$ corresponds to $T_{min}$, and is the starting point of the piecewise linear function. In this sentence, we simply explain where the slope of the piecewise linear function is changing. However, the subscript for $\kappa_{L-1}$ was incorrect and has been replaced by $\kappa_L$.

RC3#53. l.131. "Let" This sentence could be put more concisely.

We agree, this sentence has been rephrased:

"*For a GCM-RCM pair k between 1 and 20, let $Y_t^k$ represent an annual maximum for the year t between 1951 and 2100, and $T_t^k$ represent the smoothed anomaly of GMST (Sect. 2) for the corresponding GCM.*"

RC3#54. l.131. "Maximum" -> maxima.

Here, we refer to a single maximum.

RC3#55. l.131. "pair" you can loose this, the reason why you would introduce k is to denote a pair.

Thank you for this suggestion, this has been done.

RC3#56. l.133-143. "GCM$^{pair\ k}$ k is not a covariate, it's an ARBITRARY label to identify a model. "aim at adjusting the distribution of GCM-RCM pairs w.r.t. the distribution of the past observations." I cannot see where in eq. (3) the observations are involved. "adjustment coefficient" Up to here i still don't know what is an "adjustment coeff". "assume that these adjustment coefficients are constant, i.e. the same for historical and future climates." I guess this means that you have one set of coeff, not one set per piece of the piecewise lin fun. However, why not? I didn't read (Brown et al. 2014) to see if they justify this assumption. "$\mu_{GCMi}$" okay, but what is the functional form, i.e., the covariate dependence. What is the co-variate to start with? Is it the integer i?

This paragraph has been strongly modified in order to clarify how these adjustment coefficients are introduced and to explain the rationale behind this use. The adjustment coefficients are additional parameters that assume that the different members of the ensemble share the same nonstationary GEV model up to a constant shift for the location and scale parameters (see proposed illustration in Fig. 3, in response to the comment RC2#2 above). This shift can be shared by all members ("One for all GCM-RCM pairs"). In this case, there is a single parameter that represents a common shift for all members of the ensemble with respect to the observations. We also consider the possibility of having different shifts for each GCM or for each RCM ("One for each GCM","One for each RCM"),

and for all members ("One for each GCM-RCM pair"). Obviously, adding many parameters to our model is only desirable if it leads to an important improvement of our predictive skills, evaluated using the split-sample experiment. As it is not often the case in our application (see Fig. 4 of the revised manuscript), we did not test more complex forms for the adjustment coefficients. This argument thus replaces the reference to Brown et al., 2014 in the revised manuscript. Again, it must be clear that the adjustment coefficients are directly estimated by maximizing the likelihood (4) and **not** by first fitting individual GEV models to each climate simulation.

RC3#57. l.137. "because it sometimes leads to prediction failures." This is a very opaque explanation.

See response to comment RC3#24 above.

RC3#58. l.146 "all annual maxima of a given massif, i.e. annual maxima from the observations and from the 20 GCM-RCM pairs" This is the first time you give indication how you mean to use the obs and model data together.

Otherwise, i'm not at all convinced if this is any good solution to the problem that models are different from each other and from reality. You just throw reality into the mix. But if you have more and more model data, it matters less and less that you have a seed of the truth.

Thank you for this comment which is also related to the comment RC1#2 above. We agree that the combination of the observations and of the different GCM-RCM pairs in Eq. (4) is simplistic and could be improved since the information provided by the observations is diluted by the large size of the ensemble (number of members times the length of the projections). We now will discuss these limitations more clearly and in more depth in the revised paper (see the paragraph of section 6 "Conclusions and outlooks"). Different tests have been carried out in order to put more weights on the observations but the formulation of these weights was problematic. However, we see our contribution as a first step in that direction.

RC3#59. l.149. Eq. 4. We really don't need this formula. It's enough to say that you perform MLE for all the data, obs and model. Or, it would be enough, had it been worth to do, as i commented above. Furthermore, you forget to mention that you also do the fitting to individual model pairs too, or various subsets of your data.

We prefer to keep this equation that helps the reader to understand how observed maxima and simulated maxima from the projections are combined (despite the limitations of this approach). It is true that the fitting is also done for various subsets of the data for the evaluation experiments. Nonstationary GEV models are fitted to each climate simulation only for the sake of illustration in Figure 5 and it would be confusing to add this explanation at this stage of the paper in our opinion.

RC3#60. l.153-154. "the calibration of the non-stationary GEV distribution." no idea what this is.

We agree that this part of this sentence was unclear and not necessary, it has been removed.

RC3#61. l.169. Unfortunately, i do not understand what you mean by evaluating predictive performance here.

As indicated in Gneiting and Raftery (2007), the log-score is used to assign "a numerical score based on the predictive distribution and on the event or value that materializes", in our

case, the predictive distribution is obtained from the GEV distribution defined in Eq. (2) with an estimated parameter vector $\hat{\theta}$ and the value used to evaluate the predictive performance (the future data of the pseudo-observations in the model-as-truth experiment, and the observations which have been discarded from the dataset fitting in the split-sample experiment). This explanation and the reference to Gneiting and Raftery (2007) have been added to the revised manuscript as follows:

"*The logarithmic score is a proper score that can be used to evaluate the predictive performance of the fitted model (Gneiting et al., 2007).*"

RC3#62. l.171. "we select one parameterization" Rather, the selection determines the set of data you will fit.

The final GEV model is always obtained using the observations dataset and the ensemble of climate simulations composed of 20 GCM-RCM pairs. Maybe this was unclear in the original version of the manuscript and we hope that the revised manuscript clarifies this point.

RC3#63. l.171-172. "we select the number of linear pieces with a model-as-truth experiment using zero adjustment coefficients for the GEV parameters" not sure how this would be done

The assumption is that the model-as-truth experiment gives a first indication on the global evolution of the GEV parameters, based on the predictive performance of the long climate runs (1950-2100), while past observations are often limited to assess these evolutions. For this reason, most of the nonstationary GEV models fitted on past observations assume linear trends for the GEV parameters (usually the location and the scale). Our study proposes an approach to assess more complex evolutions of the GEV parameters based on a climate ensemble, even if some limitations must be acknowledged.

RC3#64. l.179. "uncertainty interval" confidence interval it's called. However, for the said reason i'm not optimistic it's any meaningful .

Thank you for this comment, we have replaced "uncertainty interval" by "confidence interval" in the revised manuscript.

RC3#65. l.181. Eq 5. On line 118 you already have this eq. This is a rather unnecessary duplication.

We agree that this equation can appear unnecessary if the reader is familiar with non-stationary GEV models. However, on l.118, the return level is provided for a stationary GEV distribution while the return level depends on *T* in Eq. 5. Furthermore, it seems important to stress that these return levels are **not** based on the adjustment coefficients.

RC3#66. l.226. Seeing the discontinuity of the slopes for many individual model pairs i wonder if the piece-wise lin model is really good. Perhaps the chi-squared test should really be done. Coles notes what to do in nonstationary EVS, which is what i also followed: https://nhess.copernicus.org/preprints/nhess-2020-117/nhess-2020-117.pdf

Thank you for this comment. First, from the comments above, it seems that there was a misunderstanding about these individual fittings which are used only for the sake of illustration. We never fit GEV models to individual GCM-RCM pairs in the rest of the paper. The gray curves in Fig. 5 showing the return levels obtained from these individual GEV models are not very smoothed indeed, maybe due to the lack of robustness of these fittings. It is in fact one important motivation for the joint inference of the likelihood (4). The goodness-of-fit of the final GEV models (colored curves in Fig. 4) have been carried out using the Anderson-Darling tests (see comment RC1#1) and these results are discussed in

the revised manuscript. Finally, we agree that an interesting approach for nonstationary GEV models is to express the GEV parameters as a linear combination of additional covariates. This is an approach that we have also considered in other applications (see, e.g. Evin et al., 2021b, for an application to extreme avalanche cycles). When there is a clear relationship between the statistical properties of the extreme variables and some climate descriptors (as is the case in the aforementioned paper between extreme cold temperatures in Europe and the arctic oscillation index), this approach is particularly powerful. However, this is not the case in our application to extreme snow loads in the French Alps where the relationships between climate indices and climate extremes related to precipitation events are often very weak (see Belkhiri and Kim, 2021). A possible idea would be to use the statistical properties of the climate simulations as covariates.

RC3#67. l.266. "100 maxima" That does not sound very many.

Our point was that 20 x 100 maxima is a large amount of information compared to the 61 observed maxima.

RC3#68. l.287 "The 90% uncertainty intervals" Why not the usual 95% but a 90% CI?

This 90% CI was preferred to rely on the 5% and 95% quantiles instead of 2.5% and 97.5% quantiles which are estimated with more uncertainties.

References

Belkhiri, L., and T.-J. Kim. 2021. "Individual Influence of Climate Variability Indices on Annual Maximum Precipitation Across the Global Scale." *Water Resources Management* 35 (9): 2987–3003. https://doi.org/10.1007/s11269-021-02882-8.

Coles, S., and L. Pericchi. 2003. "Anticipating Catastrophes through Extreme Value Modelling." *Journal of the Royal Statistical Society: Series C (Applied Statistics)* 52 (4): 405–16. https://doi.org/10.1111/1467-9876.00413.

Croce, P., P. Formichi, F. Landi, P. Mercogliano, E. Bucchignani, A. Dosio, and S. Dimova. 2018. "The Snow Load in Europe and the Climate Change." Climate Risk Management 20 (January): 138–54. https://doi.org/10.1016/j.crm.2018.03.001.

Drótos, G., T. Bódai, and T. Tél. 2015. "Probabilistic Concepts in a Changing Climate: A Snapshot Attractor Picture." *Journal of Climate* 28 (8): 3275–88. https://doi.org/10.1175/JCLI-D-14-00459.1.

Evin, G., S. Somot, et B. Hingray. 2021a « Balanced Estimate and Uncertainty Assessment of European Climate Change Using the Large EURO-CORDEX Regional Climate Model Ensemble ». *Earth System Dynamics* 12, nº 4: 1543-69. https://doi.org/10.5194/esd-12-1543-2021.

Evin, G., P. D. Sielenou, N. Eckert, P. Naveau, P. Hagenmuller, and S. Morin. 2021b. "Extreme Avalanche Cycles: Return Levels and Probability Distributions Depending on Snow

and Meteorological Conditions." *Weather and Climate Extremes*, July, 100344. https://doi.org/10.1016/j.wace.2021.100344.

Gneiting, T., and A. E Raftery. 2007. "Strictly Proper Scoring Rules, Prediction, and Estimation." *Journal of the American Statistical Association* 102 (477): 359–78. https://doi.org/10.1198/016214506000001437.

Hu, G., T. Bódai, and V. Lucarini. 2019. "Effects of Stochastic Parametrization on Extreme Value Statistics." *Chaos: An Interdisciplinary Journal of Nonlinear Science* 29 (8): 083102. https://doi.org/10.1063/1.5095756.

June-Yi Lee, Tamás Bódai. (2021) Indian summer Monsoon Variability 1st Edition, El Niño-teleconnections and beyond: Chapter 20, Future Changes of the ENSO-Indian Summer Monsoon Teleconnection, Elsevier, pp. 393-412.

Rajczak, J., and C. Schär. « Projections of Future Precipitation Extremes Over Europe: A Multimodel Assessment of Climate Simulations ». *Journal of Geophysical Research: Atmospheres* 122, nᵒ 20 (2017): 10,773-10,800. https://doi.org/10.1002/2017JD027176.

Ribes, A., S. Qasmi, and N. P. Gillett. 2021. "Making Climate Projections Conditional on Historical Observations." *Science Advances* 7 (4): eabc0671. https://doi.org/10.1126/sciadv.abc0671.

Serinaldi, F., and C. G. Kilsby. « Rainfall Extremes: Toward Reconciliation after the Battle of Distributions ». *Water Resources Research* 50, nᵒ 1 (1 janvier 2014): 336-52. https://doi.org/10.1002/2013WR014211.

Storch, Hans von, and Francis Zwiers. 2013. "Testing Ensembles of Climate Change Scenarios for 'Statistical Significance.'" *Climatic Change* 117 (1): 1–9. https://doi.org/10.1007/s10584-012-0551-0.

Tél, T., Bódai, T., Drótos, G., Haszpra, T., Herein, M., Kaszás, B., Vincze, M. (2020) The Theory of Parallel Climate Realizations – A New Framework of Ensemble Methods in a Changing Climate: An Overview, Journal of Statistical Physics, http://doi.org/10.1007/s10955-019-02445-7

Verfaillie, D., M. Lafaysse, M. Déqué, N. Eckert, Y. Lejeune, et S. Morin. « Multi-component ensembles of future meteorological and natural snow conditions for 1500m altitude in the Chartreuse mountain range, Northern French Alps ». *The Cryosphere* 12, nᵒ 4 (10 avril 2018): 1249-71. https://doi.org/10.5194/tc-12-1249-2018.

Vionnet, V., E. Brun, S. Morin, A. Boone, S. Faroux, P. Le Moigne, E. Martin, et J.-M. Willemet. « The detailed snowpack scheme Crocus and its implementation in SURFEX v7.2 ». *Geosci. Model Dev.* 5, nᵒ 3 (24 mai 2012): 773-91. https://doi.org/10.5194/gmd-5-773-2012.